# No apparent trade-offs associated with heat tolerance in a reef-building coral

Liam Lachs [1✉], Adriana Humanes[1], Daniel R. Pygas[2,3], John C. Bythell [1], Peter J. Mumby [4,5], Renata Ferrari [2], Will F. Figueira [3], Elizabeth Beauchamp[1], Holly K. East [6], Alasdair J. Edwards[1], Yimnang Golbuu[5], Helios M. Martinez[1], Brigitte Sommer [3,7], Eveline van der Steeg[1] & James R. Guest[1]

As marine species adapt to climate change, their heat tolerance will likely be under strong selection. Yet trade-offs between heat tolerance and other life history traits could compromise natural adaptation or assisted evolution. This is particularly important for ecosystem engineers, such as reef-building corals, which support biodiversity yet are vulnerable to heatwave-induced mass bleaching and mortality. Here, we exposed 70 colonies of the reef-building coral *Acropora digitifera* to a long-term marine heatwave emulation experiment. We tested for trade-offs between heat tolerance and three traits measured from the colonies in situ – colony growth, fecundity, and symbiont community composition. Despite observing remarkable within-population variability in heat tolerance, all colonies were dominated by *Cladocopium* C40 symbionts. We found no evidence for trade-offs between heat tolerance and fecundity or growth. Contrary to expectations, positive associations emerged with growth, such that faster-growing colonies tended to bleach and die at higher levels of heat stress. Collectively, our results suggest that these corals exist on an energetic continuum where some high-performing individuals excel across multiple traits. Within populations, trade-offs between heat tolerance and growth or fecundity may not be major barriers to natural adaptation or the success of assisted evolution interventions.

[1] School of Natural and Environmental Sciences, Newcastle University, Newcastle upon Tyne NE1 7RU, UK. [2] Australian Institute of Marine Sciences, Townsville, QLD 4810, Australia. [3] School of Life and Environmental Sciences, University of Sydney, Sydney, NSW 2006, Australia. [4] Marine Spatial Ecology Lab, School of Biological Sciences, University of Queensland, St. Lucia, QLD 4072, Australia. [5] Palau International Coral Reef Center, Koror 96940, Palau. [6] Department of Geography and Environmental Sciences, Northumbria University, Newcastle upon Tyne, UK. [7] School of Life Sciences, University of Technology Sydney, Sydney, NSW 2007, Australia. ✉email: l.lachs2@newcastle.ac.uk

Ocean warming is causing profound changes in marine ecosystems, and to keep pace and avoid extirpation, species must either migrate or adapt. The adaptive capacity of ecosystem engineers, such as reef-building corals, will play a disproportionately large role in the future biodiversity and function of marine ecosystems. Coral reefs continue to face unprecedented declines due to mass coral bleaching and mortality events caused by marine heatwaves[1–3]. These extended periods of anomalously high ocean temperatures are increasing in frequency and intensity under climate change[4,5]. The ability of individual corals to survive levels of heat stress sufficient to induce mass bleaching and mortality, hereafter 'heat tolerance', will likely emerge as an important trait under natural selection during the coming decades. Survivorship rather than just bleaching is a core component of heat tolerance as bleached corals can still recover and persist[6,7].

Considerable variability in coral heat tolerance exists between individuals, even within a single coral population on a single reef[8]. Currently, there are growing efforts to test novel restorative interventions, such as assisted evolution, which aims to enhance the heat tolerance of coral populations by seeding reefs with more tolerant coral colonies[9,10]. Understanding variation in heat tolerance is crucial to estimating the capacity for natural adaptation to climate change and the efficacy of assisted evolution interventions. Under both natural selection and assisted evolution, the relationships between coral heat tolerance and fitness traits (i.e., reproduction and survival—even in non-heat stress years) or other ecological traits (e.g., growth and fecundity which may or may not affect fitness) are of fundamental importance to future population persistence.

Organisms are limited by resource availability, forcing them to balance resource allocation between different physiological processes leading to trade-offs between resource-intensive traits. For instance, successful strategies to deal with drought stress are well known in long-lived birds, where less energy is allocated to reproduction in drought years to preserve cell maintenance and growth[11]. Such trade-offs will always occur between resource-limited processes. However, sometimes apparent positive associations can be found between resource-intensive traits, even when a trade-off might be expected. The variability of total resource budgets among individuals can explain this phenomenon[12], and can be associated with positive correlations across multiple traits and co-tolerance to the impacts of multiple stressors among individuals[13], despite the presence of more nuanced resource trade-offs within individuals. For instance, in oysters there is genetic evidence for trade-offs (negative correlations) between reproductive effort and both survival and growth[14], likely due to resource allocation. However, these negative correlations turn positive when these traits are measured across numerous oysters under feeding treatments[14]. This can occur due to variability in resource acquisition among individuals, which in turn can lead to differences in their total resource budgets. Subtle trait trade-offs are then easily masked by the broader population-scale energetic continuum (i.e., gradient of total resource budgets among individuals). Such positive phenotypic correlations can also be associated to genetic correlations among traits which manifest as co-tolerance of individual organisms to multiple biotic and abiotic stressors[15].

Heat tolerance in reef-building corals has been shown to have negative associations with growth[16–18], suggesting a resource trade-off. This can lead to considerable negative impacts on coral reefs at the ecosystem level[19]. For instance, *Acropora* spp. and *Pocillopora damicornis* corals dominated by thermally tolerant *Durusdinium* spp. symbiotic microalgae show considerable reductions in vital cell processes, such as carbon storage[20],

photosynthetic efficiency, and energetics[20,21]. Ultimately this results in reduced coral growth in terms of calcification rates[16,22]. Notably, this growth disadvantage can be eliminated under warming of 1.5–3 °C, as growth rates decline disproportionally with increasing temperature for corals hosting *Cladocopium* symbionts compared to those hosting *Durusdinium* symbionts[23]. The presence of mixed symbiont communities and symbiont shuffling post-bleaching can lead to flexibility in the magnitude of heat tolerance-growth trade-offs[22,24]. However, for other coral genera (e.g., *Montipora* spp.), there is mixed evidence on whether (see ref. [25]) or not (see ref. [26]) *Durusdinium* spp. symbionts (rather than *Cladocopium* spp.) influence coral host physiology and metabolism. Technological advances in photogrammetry now allow completely non-invasive determination of colony growth. This has some specific logistical advantages for repeated monitoring of corals in the field compared to other techniques which require removing corals from the substrate (e.g., buoyant weight) or causing potential harm (e.g., linear extension using staining)[27].

Many coral populations are dominated by a single symbiont taxon or a single symbiont community type[28,29]. In these cases, do trade-offs between heat tolerance and other traits persist? Recent genomic evidence based on corals from contrasting thermal environments suggests that the shift in allele frequencies associated with coral host-derived heat tolerance are often associated with a fitness cost[28]. However, it is yet to be tested whether trade-offs between heat tolerance and other ecological traits exist for corals that share the same Symbiodiniaceae community. Considering their prevalence in numerous other taxa, including Crustacea, Insecta, and Chordata[30–32], it is likely that heat tolerance-related trait trade-offs also affect coral hosts. As coral adaptation occurs locally, not globally, and since endosymbiont communities are relatively uniform across local scales, it is important to resolve the extent of host-derived heat tolerance trade-offs to better predict coral adaptation to climate change.

Egg and sperm development are resource intensive processes and have been suggested as potential costs to growth and heat tolerance[33]. It is reasonable to expect trade-offs between heat tolerance and fecundity given the evidence for heat tolerance-growth trade-offs in corals, and the fact that growth and gamete production are both resource-intensive processes. Evidence has shown that temperature stress can reduce hard and soft coral fecundity (i.e., egg density and volume)[34,35], and has suggested that corals may reabsorb their oocytes to divert energy away from reproduction and into growth under certain types of stress such as fragmentation[36,37]. However, there has not yet been an assessment of the relationship between fecundity and heat tolerance to test for associations or trade-offs which could be crucial to understand population fitness and performance.

Here, we tested for ecologically relevant associations and trade-offs between heat tolerance and three ecological traits in a common species of Indo-Pacific coral: colony growth, fecundity, and symbiont community composition. To measure these traits we combined: (i) a long-term (*sensu*[38]) 5-week marine heatwave emulation experiment to measure heat tolerance; (ii) interannual comparisons of 3D models of individual coral colonies to measure growth (change in live surface area and colony volume); (iii) polyp counts and dissections to measure fecundity; and (iv) ITS2 sequencing to determine Symbiodiniaceae community composition. We employ Bayesian methods for solving simple trait trade-off linear regressions (in the form: heat tolerance ~ $\beta_0 + \beta_1 \times$ trait + error) to allow the quantification of uncertainty via inspection of posterior distributions, specifically testing the odds of no trade-off occurring (i.e., $\beta_1$ slope value > 0).

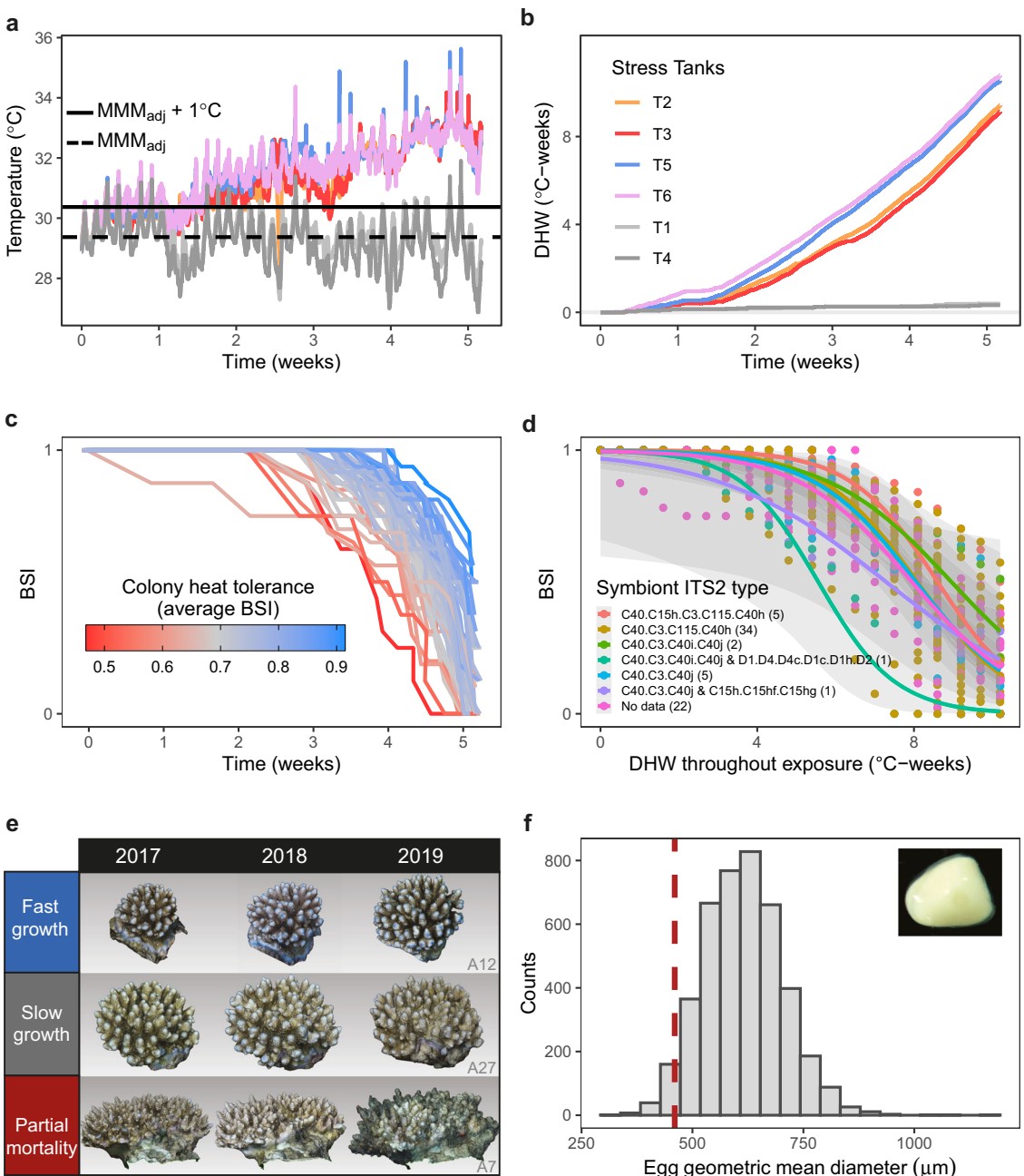

**Fig. 1 Univariate exploration of traits among *Acropora digitifera* colonies showing marine heatwave emulation, 3D colony growth, and fecundity. a** The experimental marine heatwave exposure conducted on 6 fragments per colony lasted 5 weeks reaching approximately +3.5 °C above the local climatological baseline (MMM$_{adj}$). Colour legend is shared with panel **b** with heated tanks and procedural control tanks (T1 and T4). **b** This translated to accumulated degree heating weeks (DHW) of ~10 °C-weeks. **c** Bleaching and mortality responses (BSI) of each colony (individual lines) are shown throughout the experiment with a horizontal jitter to separate overlapping lines. **d** The BSI-DHW relationship (where BSIs are corrected for DHW drift among tanks) was unaffected by symbiont ITS2 type, showing number of colonies with each symbiont ITS2 type in brackets. **e** Interannual comparisons of coral colony structure-from-motion 3D models revealed marked variability in growth rates, with examples from some common growth types shown here, and all other colony models shown in Fig. S4. **f** Fecundity measurements from 2 fragments per colony match closely to previous estimates of egg diameter for *Acropora* spp. (red dashed line)[39].

## Results

**Heat tolerance variability**. Within the studied coral population, we found substantial variability in coral heat tolerance measured as bleaching and mortality responses throughout a marine heatwave emulation experiment. At the beginning of the experiment all colonies had all replicate fragments healthy, corresponding to BSI (bleaching and survival index) values of 1. The final heat stress exposure reached a DHW (degree heating weeks) of

10.7 °C-weeks (Fig. 1a, b), a level which would likely induce a mass bleaching and mortality event in nature. By this final exposure, 47% of colonies had all replicate fragments dead (BSI = 0), while most remaining colony fragments were bleached (25% of all fragments), translating to BSI values < 0.56 (Fig. 1c). Meanwhile, in the unstressed procedural control tanks, all representative fragments from each colony remained alive, showing that the experimental setup (aquarium lights, flow

through filtered water etc.) was suitable for keeping fragments alive and that heat stress was the cause of bleaching and mortality responses (i.e., rather than effects due to the tank setup). Coral heat tolerance throughout the experimental exposure, expressed as the average BSI, ranged from 0.47 to 0.91 among colonies. The critical level of accumulated heat stress (DHW—degree heating weeks) at onset of bleaching and mortality (first fixed DHW value at which BSI ≤ 0.75) was highly variable across colonies, with an average critical DHW of 6.7 °C-weeks (±1.1 °C-weeks SD) but ranging from 4.3 to 9.2 °C-weeks in the least and most tolerant individuals. Notably, this range corresponds to almost an entire categorical shift of the NOAA Coral Reef Watch bleaching alert system (e.g., Alert Level 1 to 2, moving from mass bleaching expected to mass mortality expected).

**Similar Symbiodiniaceae communities.** Symbiodiniaceae community composition was consistent among colonies, whereby all individual colonies were dominated by C40 *Cladocopium* spp. (Fig. 1d) and 96% of colonies contained a single ITS2 type profile. For 70% of colonies, the dominant symbiont strain was C40-C3-C115-C40h. Only 2 colonies had mixed symbiont communities, with 29% relative abundance of C15h-C15hf-C15hg in colony A75 and 31% D1/D4-D4c-D1c-D1h-D2 in colony A96. There was no significant effect of symbiont ITS2 type on bleaching and mortality responses (Fig. 1d) with strongly overlapping confidence intervals across all ITS2 type profile groups (Fig. S1) (GLMM Tukey test, $P > 0.05$ for all pairwise comparisons, Table S2). Although the colony with *Durusdinium* spp. symbionts appeared to bleach and die faster than other colonies, with $N = 1$ for this ITS2 profile type there was insufficient statistical power to detect whether this particular BSI trajectory differed from the rest of the population. This was also shown from principal component analysis based on ITS2 type profiles and grouping colony heat tolerance into broad equal-sized categories (average BSI: high ≥ 0.8 > medium ≥ 0.7 > low; Fig. S2), with strong overlap among each category.

**3D colony growth.** Coral colony growth was highly variable in terms of both live surface area growth ($216 \pm 722$ cm$^2$ yr$^{-1}$, average ± SD) and volumetric growth ($404 \pm 647$ cm$^3$ yr$^{-1}$, average ± SD). There were a range of relatively fast-growing colonies, stable-sized colonies, and others that either lost live tissue or volume through partial mortality or breakage (example colonies of each growth type shown in Fig. 1e, and all colony 3D models shown in Fig. S4). Once growth metrics were corrected by initial colony size, there was no significant trend between live surface area growth and colony size (linear regression, $P > 0.05$; Fig. S6). However, there was a significant negative trend between total volumetric growth and colony size, whereby larger corals had lower rates of volumetric change (linear regression, $P = 0.042$; Fig. S6). Notably, colony partial mortality had no effect on fecundity or symbiont community traits (ANOVA and Tukey tests or pairwise Bonferroni-corrected Wilcoxon tests, all $P$ values > 0.05; Fig. S7), a weak negative effect on heat tolerance (average BSI; ANOVA, $P = 0.023$; Fig. S7), and a stronger negative effect on growth which was expected as tissue loss reduces size (pairwise Bonferroni-corrected Wilcoxon tests for live surface area growth, $P = 0.006$; ANOVA and Tukey tests for volumetric growth, $P = 0.005$; Fig. S7).

**Fecundity.** Egg density averaged 4.9 eggs per polyp (±1.4 eggs per polyp SD) across all coral colonies, with polyps containing between 1 and 11 eggs. The geometric mean diameter of eggs was in line with that shown for *Acropora digitifera*[39], although slightly larger (Fig. 1f, red dashed line). Average egg volume across all

colonies was $0.11 \pm 0.03$ mm$^3$ SD, ranging from 0.06 to 0.2 mm$^3$, similar to that shown in previous work[34]. The estimates of total colony egg production ($274,590 \pm 236,624$ eggs colony$^{-1}$, average ± SD) and total colony egg volume ($32 \pm 27$ cm$^3$ colony$^{-1}$, average ± SD) were highly variable among colonies, likely due to variability in colony size.

**Lack of trade-offs with heat tolerance and positive trait correlations.** There was no evidence for trade-offs between heat tolerance (average BSI per colony) and either growth (Fig. 2a), fecundity (Fig. 2b), or Symbiodiniaceae community composition (Fig. S1, Fig. S2). The lack of a relationship with Symbiodiniaceae relates to the high similarity of the symbiont community among colonies. Contrary to expectations, we found positive associations between heat tolerance and growth metrics (live surface area growth and volumetric growth). Despite having a weak association due to high levels of uncertainty (i.e., 95% credible intervals of the slope posterior distributions intersected zero, Fig. 2a), the probability that these slopes were positive, and that growth and heat tolerance act in concert was high; 90% probability for live surface area growth and 94% probability for volumetric growth (Fig. S8). Although the slope values seem small (i.e., $\times 10^{-5}$) this is due to a disparity in the order of magnitude between growth values ($\times 10^3$) and average BSI values ($\times 10^{-1}$). Accordingly, these differences in growth correspond to weak but potentially important shifts in average BMI. For instance, moving from the 10th to 90th percentile of volumetric colony growth corresponds to a shift in heat tolerance from the 40th to 60th percentile of the population (Fig. 2a), or an increased bleaching heat stress tolerance of 0.7–0.9 °C-weeks (Fig. S9). These results were markedly similar even when colonies that had experienced shrinkage (reduction in surface area or volume) were excluded from the analysis (Fig. S10). In comparison, any effect of fecundity metrics (total colony egg production and total colony egg volume) on heat tolerance were unmeasurable given our sampling design (Fig. 2b), with close to 50:50 odds of the relationship being negative or positive (trade-off or co-benefit) (Fig. S8). This trend remained the same even when the most fecund colony, a potential outlier, was removed from the analysis.

Throughout the course of the 5-week heatwave emulation experiment (Fig. 3), the relationships between coral colony growth (measured from 3D models on the reef) and instantaneous bleaching and survival responses (measured as BSI) were variable. The onset of bleaching responses occurred after ~3 weeks of elevated temperatures (Fig. 1b) at a DHW value of ~4–6 °C-weeks (Fig. 3a, Fig. S9). However, bleaching onset was delayed in fragments sampled from positive-growth colonies compared to negative-growth colonies by approximately 1 °C-week (Fig. 3a, Fig. S9). Accordingly, during this period of differential bleaching onset (at ~5 °C-weeks), the slope of the BSI-growth relationship (concept shown in Fig. 3b) was strongly positive showing a pronounced peak (e.g., 95% credible interval of the slope not intersecting zero, Fig. 3c). Faster growing colonies needed higher levels of heat stress to trigger the onset of bleaching responses.

The BSI-growth slope progressed throughout the experiment with a double peak pattern. Moving on to 7 °C-weeks, as bleaching responses aligned across all colonies of different growth rates, the BSI-growth slope decreased from the first bleaching-associated peak toward zero (Fig. 3c). As the heat stress exposure reached 9 °C-weeks, the BSI-growth slope again increased to a second peak, this time reflecting the delayed onset of mortality responses in positive-growth colonies compared to negative-growth colonies. Toward the end of the exposure (10.7 °C-weeks),

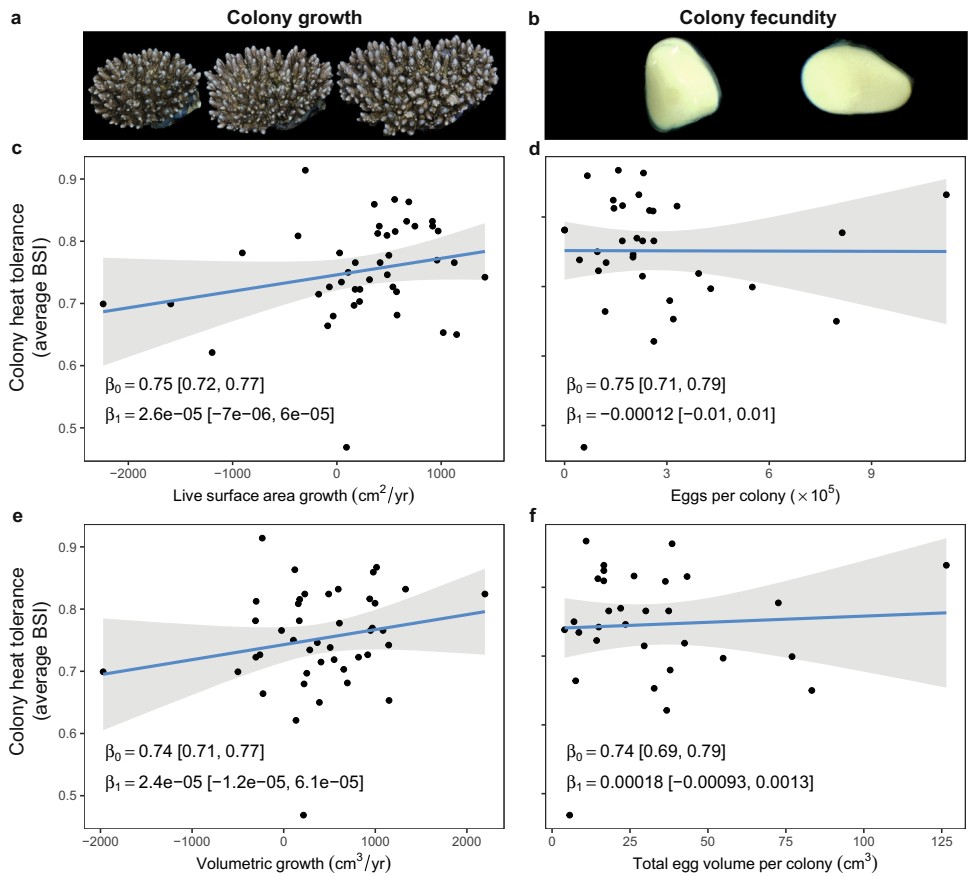

**Fig. 2 No apparent trait trade-offs associated with overall coral colony heat tolerance.** Associations between colony heat tolerance (average BSI through time) and either corrected growth metrics (**c** – live surface area growth, and **e** – volumetric growth) or colony fecundity (**d** – eggs per colony, and **f** – total egg volume per colony). Example 3D models used to measure growth are shown for a single colony (**a**), and microscope images of eggs which were counted and measured to determine fecundity are also shown (**b**). Each plot shows the median and 95% credible interval of intercept ($\beta_0$) and slope ($\beta_1$) parameters of linear regressions, where a negative slope would reflect a trade-off, however weak positive associations are present for growth (**c**, **e**).

the BSI-growth slope moved back toward zero, reflecting critical bleaching and mortality responses across all colonies regardless of colony growth rates.

Notably, beyond 7 °C-weeks, the uncertainty around the BSI-growth slope widened (Fig. 3c), intersecting with zero. Yet despite this, the probability of a positive BSI-growth slope (i.e., no trade-off) remained at 72–91% throughout the later stages of the heat stress exposure as bleaching and mortality responses progressed to critical levels.

The strength of this temporal analysis (Fig. 3) is that it is based on raw BSI values (not summary statistics like average BSI). While the analysis of average BSI (Fig. 2), suggests a weak co-benefit between heat tolerance and growth but with high uncertainty, the temporal analysis (Fig. 3c) shows the nuance of this co-benefit. It is at the onset of bleaching and at the onset of mortality that there is a maximum co-benefit (greatest slope value), and at the onset of bleaching that there is the highest level of confidence of this co-benefit (95% credible intervals not overlapping zero).

Throughout the heat stress exposure, the progression of instantaneous BSI-growth relationships for volumetric growth (Fig. S11) showed a similar pattern to that of live surface area growth (Fig. 3). Slopes were consistently positive with credible intervals deviating from zero at bleaching onset (Fig. S12). However, the progression of instantaneous BSI-fecundity relationships (total colony egg production, and total colony egg volume) was static, centred at zero throughout the heat stress exposure (Fig. S11). These results show that growth has a positive

association with heat tolerance, but that fecundity has no association with heat tolerance (Fig. S12).

## Discussion

In coral reef ecology, theory and evidence suggest that high coral heat tolerance is associated with a growth trade-off, especially for corals hosting certain symbiotic dinoflagellates (e.g., *Durusdinium trenchii*)[16,19,21,23]. Here, we investigated heat tolerance-associated trait trade-offs in a shallow outer reef crest coral population primarily hosting C40-dominated Symbiodiniaceae communities. We found no evidence for trade-offs between coral heat tolerance and either growth or fecundity.

Contrary to expectations, we found weak positive associations between heat tolerance and colony growth. Indeed, during a marine heatwave emulation experiment, fragments taken from faster growing coral colonies on the reef were able to withstand higher levels of experimental heat stress before the onset of bleaching and mortality. Previous work has identified considerable trade-offs between heat tolerance and growth in terms of calcification rates caused by the presence of different symbionts[16], flagging this as a potential barrier to successful coral adaptation under climate change[19]. Our results show that heat tolerance and whole colony growth (in terms of surface area and volume) can be positively associated, offering a more optimistic outlook for coral populations. This builds on the recent finding of co-tolerance of individual corals to multiple stressors (e.g., thermal stress, bacterial infection)[15]. Particularly we found a double-peak pattern in BSI-growth regression slopes throughout a 5-week heat stress

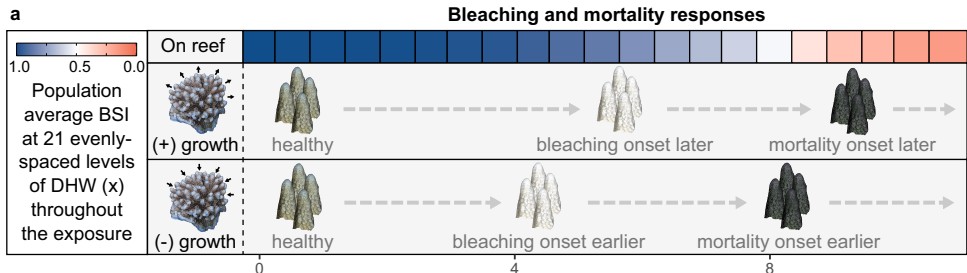

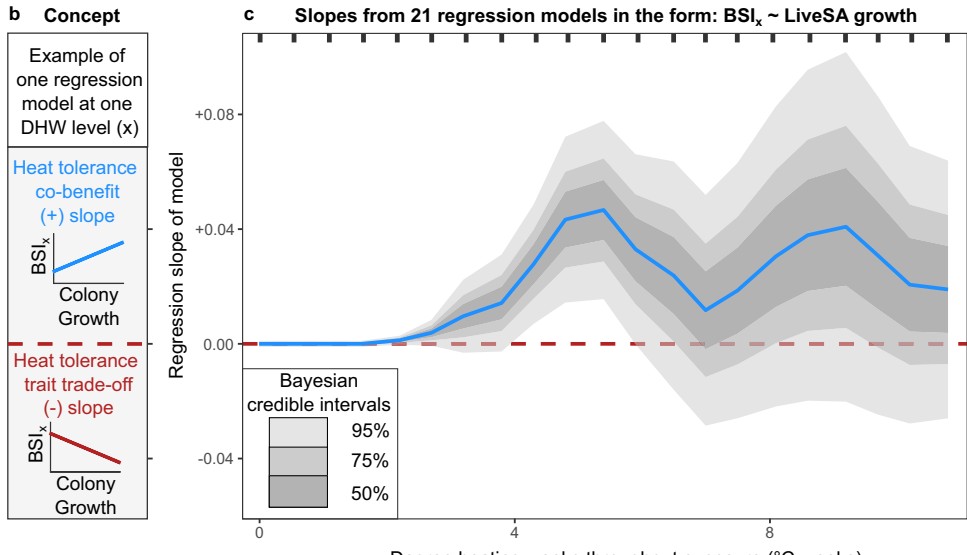

**Fig. 3 Progression of bleaching and mortality throughout heat stress exposure as a function of donor colony growth rate.** The association between corrected annual colony growth in terms of live surface area (LiveSA from 3D models on the reef) and acute bleaching survival responses (BSI) of colony fragments in heat stress tanks. **a** Progression of bleaching and mortality responses (colour bar) is shown at 21 evenly spaced degree heating week (DHW) levels (x) on the *x*-axis. Fragments from positive-growth colonies take longer to bleach and die than those from negative-growth colonies. **b** Conceptual diagram linking the slope of one BSI-growth regression model (for all colonies with available data) at a single DHW level (x) to positive trait correlations (positive slope) or trade-offs (negative slope). A slope of zero is shown by the dashed red line. **c** Progression of the relationship between colony growth and fragment BSI at 21 DHW levels (x, bold ticks on top), shown as the median slope (blue line) and the 50%, 75%, and 95% Bayesian credible intervals (grey shading) from posterior distributions of 21 regression models.

exposure (Fig. 3c). First the onset of bleaching was delayed in fragments of faster growing coral colonies (Fig. 3c, first peak), and then the onset of mortality was delayed (Fig. 3c, second peak). Our findings suggest that selecting corals for heat tolerance either through natural selection (i.e., selective mortality of heat sensitive individuals) or assisted evolution (i.e., propagating heat tolerant individuals) may not compromise growth or fecundity.

A key consideration here is separating the distribution of traits within a contemporary population from what happens to the future population as temperatures continue to rise under climate change. In general, we found that more heat tolerant individuals also tended to have higher colony growth rates. This implies that post-bleaching coral populations may not necessarily have lower overall growth in terms of changes in colony size. However, the existence of thermal optima—as demonstrated for coral calcification[40] and photosynthesis[41]—still imply that long-lived corals may experience declines in their growth as temperatures rise, even if they are the more heat tolerant members of the earlier population. As we did not measure growth post-bleaching, there is a need for future research to understand the plasticity of trait associations after stress. As such, further work is also needed to understand whether selection for heat tolerance can also select for other beneficial traits. Our study focussed on corals from similar

depths on a single reef to limit the influence of the environment on organism physiology. Further research is needed to understand if positive associations between heat tolerance and growth are also present in other coral species and over broader spatial scales.

Under climate change, coral heat tolerance will likely be one of the most important fitness-related traits in determining population persistence or collapse[42,43]. However, heat tolerance can come at a cost to other traits, like growth. This premise is typically based on Symbiodiniaceae-derived coral heat tolerance, where certain dominant symbiont taxa (e.g., *Durusdinium tren-chii*) confer higher tolerance at the expense of photosynthetic energetics and ultimately growth as calcification[44]. Many coral populations, particularly in the Indo-Pacific, host Symbiodiniaceae communities that are either dominated by a single taxon of symbiotic algae or by a single community type with similar relative abundance of different symbiont taxa[28,45]. Determining whether heat tolerance trade-offs persist for corals hosting broadly similar symbiont communities can improve our understanding of coral population functioning and the potential of adaptation to climate change.

Trade-off theory suggests that corals have contrasting strategies, either being resistant to high temperatures or showing

enhanced calcification rates, with concurrent disadvantages being reported for both strategies[16,17]. Our results show that heat tolerance can be positively associated with whole colony growth, where some individuals would expectedly have higher fitness due to excelling in multiple traits simultaneously. However, our results cannot be compared directly to calcification-based studies since we have measured growth as changes in colony size (to capture net positive and negative changes) which may bear different implications for coral populations. Still, organisms must partition resources among costly physiological processes, suggesting that one should find trade-offs between colony growth and heat tolerance. However, total resource budgets can be highly variable among individuals, especially in wild populations, due to processes such as the efficiency of resource acquisition from the environment[12]. As such, even though trade-offs must occur at some level of biological organisation, they can be masked at ecological scales due to high variability of resource acquisition among individuals. In line with trade-off theory, such energetic variability among individuals can result in apparent positive associations among resource intensive traits[12,14]. It is likely that the physiological processes underpinning high growth rates are also linked to high resilience to heat stress, as suggested by the concept of co-tolerance[13]. None of the colonies with high live surface area growth rates underwent partial mortality. Those with colony shrinkage due to partial mortality were likely to have been affected by other stressors, such as disease, competition, physical damage, or predation, and as such were also associated with lower levels of heat tolerance. Yet even when colonies that had experienced shrinkage (reduction in surface area or volume) were removed from the analysis a weak positive association between heat tolerance and colony growth remained, suggesting the trend observed in this study was not an artefact of colonies undergoing shrinkage (e.g., through processes including predation, tissue necrosis, or breakage). The physiological cost of tissue repair, fighting infection or regrowth after breakage could deplete energy reserves[46,47] rendering corals more susceptible to bleaching and mortality under acute heat stress. It may be possible that while we find weak positive associations between heat tolerance and whole colony growth, trade-offs with other traits such as calcification could still exist[16]. Such a trade-off could compromise individual fitness of more heat tolerant corals particularly during storm surges when there is a higher risk of colony breakage.

Tolerance to extreme temperature stress is a vital trait for corals in the weeks or months during marine heatwaves. However, heatwaves currently do not happen every year and generally occur only in the warmest months. As such, heat tolerance is unlikely to directly benefit corals during cooler months or years, without considering associations between heat tolerance and other traits. Comparatively, other traits like growth or fecundity are of importance throughout every year in sexually mature adult corals (i.e., over 3 years old for *Acropora* spp.[48]). Colonies grow year-round and typically spawn during one season per year[49], whilst developing eggs for the rest of the annual gametogenic cycle. Together, these results suggest that corals exist along an energetic continuum, where positive trait correlations may be derived from underlying physiological drivers like immunity[47], feeding efficiency[50], or energy storage[51]. Energetic variability could then result in higher levels of fitness and better performance across suites of different traits. Weak positive associations with heat tolerance occurred with growth but not fecundity, suggesting that the drivers of energy allocation to fecundity may act independently of heat tolerance.

A trade-off between coral heat tolerance and key ecological traits like growth would have considerable negative implications for natural evolution under climate change[19]. This would also apply to restoration efforts involving assisted evolution that aim to boost population resistance to heat stress by propagating more heat tolerant coral individuals via selective breeding or assisted gene flow. If coral heat tolerance was associated with lower growth or fecundity for instance, then out-planting large numbers of corals with these traits would have potentially damaging effects on natural population fitness. However, we found no evidence for such trade-offs between heat tolerance and either colony growth or fecundity, for a coral population associated with the same Symbiodiniaceae community. Although further work will be needed to understand whether these trends persist across larger spatial scales and for other species, our results suggest that selecting corals for heat tolerance through either natural selection or assisted evolution is unlikely to come at a cost to growth or fecundity. Under climate change, coral heat tolerance will be increasingly important to the persistence of coral populations. Contrary to expectations, selection for heat tolerance may not necessarily compromise other important parts of the coral life-history.

## Materials and methods

**Model system**. To minimise the influence of environmental and interspecific drivers, we measured variation in traits on a single coral population at Mascherchur reef, Palau, Micronesia (7° 17′ 29.3″ N, 134° 31′ 8.0″ E), a semi-sheltered outer reef crest. The shallow-water Indo-Pacific reef builder, *Acropora digitifera*, was chosen as a model species due to its high local abundance, broad geographic distribution, and corymbose growth form that allows sub-sampling of branches (fragments) to provide intra-organism statistical replication without sacrificing the colony. Seventy coral colonies were tagged at 2–3 m depth and surveyed in situ repeatedly for different traits between 2017 and 2019. Large adult colonies of similar diameter (24 ± 8 cm, average ± SD) were chosen to limit the size-related variability in total resource budgets among individuals which could obscure trade-off relationships[12].

**Marine heatwave emulation experiment**. Colony heat tolerance was determined in August 2018 at the Palau International Coral Reef Center by subjecting replicate fragments of each colony to a long-term 5-week marine heatwave emulation experiment (all tank experiment details are given in Table S1). In comparison to short heat-shock experiments that typically last 1–2 days, this experimental temperature profile was designed to match more closely the duration of natural marine heatwaves[8,38], with the assumption that the phenotypic bleaching and mortality responses would be more ecologically relevant. After fragments were collected and given a 7–10-day acclimatisation period under ambient thermal conditions in all tanks, temperature was increased gradually (~0.8 °C week⁻¹, Table S1) over the time course of the experiment (35 days), reaching a final bleaching-level temperature of approximately 33 °C, or 3.5 °C above the local climatological baseline (MMM – maximum of monthly means, Fig. 1a, Table S1, see ref. [8] for more detailed tank setup description). The use of flow-through tank systems allowed an element of natural diel temperature variability[52] in all tanks (4 heat stress and 2 procedural control tanks), while aquarium lights provided light conditions during the acclimatisation period and the heat stress exposure at a daily average intensity of 400 µmol m⁻² s⁻¹, corresponding to the average light intensity measured in Mascherchur at midday[8]. HOBO loggers, with 0.14 °C resolution and 0.45 °C accuracy, were calibrated against a RBR TR-1050 using the average offset for temperatures between 27 and 35 °C in increments of 0.5 °C. Calibrated HOBO loggers were placed in each tank and recorded temperatures at 10-min intervals. To relate coral bleaching and mortality responses to accumulated heat stress, not instantaneous temperature, we calculated heat stress for each tank using the Degree Heating Weeks (DHW) metric. DHW was developed by the National Oceanic and Atmospheric Administration's Coral Reef Watch to provide a real-time coral bleaching alert system based on satellite-derived sea surface temperatures. DHW is a daily measure of both the intensity and duration of heat stress, calculated by accumulating temperature anomalies > 1 °C relative to a climatological baseline (MMM) over a 12-week (84-day) rolling window[53]. To allow for comparisons between the DHW from our experiment and the NOAA bleaching forecasts, we adjusted the MMM from the 5 km grid cell encompassing the collection site based on the relationship between daily time series of 5 km sea surface temperature (CoralTemp v3.1) and daily averaged in situ temperature (recorded from additional HOBO loggers at the collection site;[8]), producing the local climatological baseline (MMM$_{adj}$ – adjusted MMM). This builds upon the eDHW method which suggest using the satellite-based MMM to compute experimental DHWs[54]. However, our previous work on Mascherchur reef has found that the eDHWs underestimate true DHWs due to a mismatch between the satellite data and in situ reef conditions[8]. Notably the NOAA CRW bleaching risk forecast considers DHW of 4 and 8 °C-weeks as Alert Level 1 (significant bleaching expected) and Alert Level 2 (significant bleaching and mortality expected), respectively[53]. The final accumulated heat stress exposure reached in this experiment was 10.7 °C-weeks.

**Heat tolerance**. Fragments (6 per colony) were dispersed in random locations among the four heat stress tanks (4 fragments colony$^{-1}$) and two procedural control tanks (2 fragments colony$^{-1}$). In total, fragments from 66 of 70 colonies were exposed to the assay, as four colonies were not found during collection. If fragments from a colony died in the procedural control tank which was under non-stressful ambient temperature conditions, it was an indication of handling effects for that colony, so all remaining fragments from the colony were removed from the experiment and the colony was not assigned a heat tolerance score (2 colonies). The health status of each fragment was scored visually into five categories based on stark whiteness and tissue state (see below) at intervals of between 1 and 3 days (total of 16 timepoints over 35 days). Notably the bleaching scores were highly correlated with pigment concentration and symbiont density[8]. We used a bleaching survival index (BSI), the inverse of the commonly used bleaching and mortality index (BMI), to categorise coral bleaching and survivorship responses[55]. This was done in order to have a positive correlation between the bleaching survival index and heat tolerance. For BSI, $c_1$ to $c_5$ are the proportion of replicate fragments (per colony) recorded as healthy ($c_1$), half bleached ($c_2$), bleached ($c_3$), partial mortality ($c_4$), or dead ($c_5$), and $N$ is the total number of categories (here $N = 5$). For example, a colony whose replicate fragments are either all healthy or all dead, will have a BSI value of 1 or 0, respectively.

$$\text{BSI} = 1 - \text{BMI} \tag{1}$$

$$\text{BMI} = \frac{0c_1 + 1c_2 + 2c_3 + 3c_4 + 4c_5}{N - 1} \tag{2}$$

Here we define the onset of the bleaching and mortality response to occur when BSI declines below 0.75. This BSI score was chosen as it represents a colony with an average health status of partially bleached across all replicate fragments, a health status indicating that bleaching and mortality responses have started and will progress further[8].

To remove potential biases relating to lagged DHW profiles among tanks, we followed the method outlined in full detail in ref. [8] which aligns DHW profiles among tanks and interpolates health status scores at fixed DHW values with fixed intervals, providing unbiased BSI values among colonies. As the BSI of a particular colony is an instantaneous measure and will change throughout the heat stress exposure, the colony's overall heat tolerance was calculated as the average BSI through time.

**Symbiont identification**. The composition of the symbiont community was identified from one tissue scraping (<1 cm) per colony sampled in March 2018 (September 2018 for two colonies not found in March), and is assumed to be stable since no major disturbances occurred during this time period[29]. Due to some colonies not being relocated during specific surveys, DNA samples were available for 51 colonies. Polymerase chain reaction (PCR) was used to amplify DNA extracted from coral tissue and then sent for ITS2 sequencing. The hyper conserved ITS2 region was chosen to facilitate integration with the SymPortal database which assigns symbiont ITS2 Type profiles or distinct intragenomic variants (DIVs) that represent different symbiont taxa[56]. A full description of the DNA extraction, sequencing protocol and Symportal analysis are provided in the supplementary materials (S1).

**Size and growth metrics**. Previous studies investigating growth-heat tolerance trade-offs in corals have measured growth as calcification rates based on Calcium incorporation[40] or buoyant weight techniques[16], which are both able to detect changes in total CaCO$_3$ growth including skeletal density and secondary skeletal infilling. These methods are invasive and require manipulation of the coral colony, which can potentially influence coral fitness[27]. Moreover, as colonial organisms, it is possible for corals to experience shrinkage and still survive (i.e., partial mortality or reduction in size) and shift between net positive or negative growth over multiple occasions. This phenomenon may be undetected using some growth measurement techniques (e.g., buoyant weight). Therefore, at the colony level we deemed it more appropriate to use photogrammetry, a non-invasive method of measuring growth that can capture any changes in colony size with high accuracy and precision[57,58].

Coral colony size was measured from 3D models at successive timepoints throughout the study (November 2017, May 2018, and February 2019). Overall, 114 3D reconstructions were built for 45 coral colonies ($n = 24$ for all time points, and $n = 21$ for two time points). Following Ferrari et al.[57], Metashape Professional (v 1.5, Agisoft) was used to construct 3D surface meshes using structure from motion photogrammetry, which had >93% photo alignment on average and an average (±SD) resolution (distance between mesh vertices) of 1.1 (0.4) mm and average (±SD) scaling error of 0.36 (0.38) mm. Geomagic Control (v 2015, 3D Systems) was used to align successive coral colony models to give a common base, and compute three size metrics: live surface area (LiveSA, excluding dead regions without live tissue), total surface area (SA) and total volume (V). The initial maximum diameter (D) was recorded in situ in 2017 for each coral colony using a measurement tape (i.e., independently of the 3D models). Further details of photogrammetry methods can be found in the supplementary materials (Supplementary Text 2, Fig. S3, Table S3).

Annual growth rates, in terms of live surface area and volume, were estimated as the change in size relative to the time interval between successive photogrammetry surveys. For those colonies with two size comparisons (i.e., 2017–2018 and 2018–2019), the average of both was taken. Due to some colonies not being relocated during specific surveys or model building issues, photogrammetry model comparisons were available for 45 colonies. However, variation in initial colony size can introduce bias to raw areal or volumetric growth measurements (i.e., for identical linear extension rates, large colonies appear to grow faster than small colonies in terms of surface area and volume). Therefore, growth metrics must be corrected for colony size. One option is to calculate percentage growth, by dividing the growth rate by the corresponding initial colony size. However, this results in growth overestimation for small colonies that may easily double in size from one year to the next (i.e., growth rate of 100%). Another option, which we adopted here, is to multiply the growth rate by a dimensionless adjustment factor (AF) that represents the difference between the population mean colony size and the size of an individual colony. AFs for each colony (i) were calculated as the ratio between the average initial size across all colonies relative to the initial size of colony i, such that

$$\text{Areal\_AF}_i = \frac{\bar{D}}{D_i} \tag{3}$$

$$\text{Volumetric\_AF}_i = \frac{\bar{A}}{A_i} \tag{4}$$

To avoid further bias, the colony size metrics used for the AF calculation were derived from D independently of the 3D models and were one dimension lower than their corresponding growth metric (e.g., correct the SA growth rate based on the diameter). For three colonies that lacked empirical measurements of D but had 3D photogrammetry models available, D was predicted from the initial SA of the photogrammetry model, such that $\log(D) = -0.738 + 0.537 \times \log(\text{initial SA})$ ($F_{40/1} = 239.1$, $R^2 = 0.86$, $P < 0.001$) (Fig. S5). The initial area (A) for use in the volumetric AF was given by the product of D times initial height (assumed to be D/2). Size-dependencies of each metric were tested using linear regressions. Notably, partial mortality was recorded from field surveys categorically, such that a colony without partial mortality must have been healthy continuously from the first until last 3D photogrammetry survey.

**Fecundity**. Fecundity was measured as egg density (number of eggs per polyp) and egg volume (average egg volume per polyp). Two fragments from each tagged coral colony in the reef were removed prior to spawning in March 2018 or 2019, decalcified using 10% hydrochloric acid and stored in ethanol. Due to some colonies not being relocated during specific surveys or sample preservation issues, polyp fecundity measurements were available for 47 colonies (25 in 2018 and 22 in 2019). Ten polyps from each fragment were dissected using a dissection scope and each egg was photographed alongside a scale using an attached digital camera. All image analysis was performed using the semi-automated SizeExtractR workflow[59] to annotate polyp images and compute egg density and volume from the 20 polyps per colony. Egg volume (EV) was estimated from the geometric mean diameter (GMD)[59]. We then estimated total colony egg production (TEP) and total colony egg volume (TEV) by combining dissection-derived fecundity data (EC – egg counts per polyp – and EV, respectively) with per colony estimates of polyp density per unit area (PD) and 3D model-derived estimates of total live surface area (LiveSA).

$$\text{EV} = \frac{4}{3} \times \pi \times \left(\frac{\text{GMD}}{2}\right)^3 \tag{5}$$

$$\text{TEP} = \text{EC} \times \text{PD} \times \text{LiveSA} \tag{6}$$

$$\text{TEV} = \text{EV} \times \text{TEP} \tag{7}$$

**Statistics and reproducibility**. Traits hypothesised to influence coral heat tolerance (average BSI) were colony growth (in terms of both LiveSA and V), colony fecundity (total egg production and total egg volume), and symbiont community. Trade-off analyses were conducted by regressing predictor variables against colony heat tolerance using Bayesian general linear models. In these regressions a negative slope represents a trait trade-off while a positive slope shows positive correlations among traits. Bayesian models were used to facilitate a more intuitive interpretation of model uncertainty[60,61] and fit using integrated nested LaPlace approximation in the statistical package R-INLA[61,62]. Uncertainty around a model parameter (e.g., the regression slope) is commonly described from using the lower and upper 95% credible intervals, which bound 95% of the area under a posterior distribution density curve. To calculate the probability of no trade-off between heat tolerance and other traits (i.e., a flat line or even a positive association with a slope greater than zero), we measured the proportion of the posterior distribution which exceeded zero. The effect of multivariate symbiont community type on sigmoidal BSI-DHW responses was tested using generalised linear models with a binomial response distribution and post-hoc Tukey tests. In addition, confounding effects of partial mortality on all traits were tested using ANOVA paired to post-hoc Tukey tests or non-parametric equivalents.

During a long-term heat stress exposure, the bleaching and mortality response of sampled fragments may vary through the time course of the experiment. Therefore, we were interested to test not only the relationship between overall heat tolerance (BSI averaged across the whole heat stress exposure) and other traits, but also the relationship between the instantaneous BSI based on replicate fragments (response variable) and the known colony-level traits (predictor variable, e.g., fecundity or colony growth). To link bleaching responses to their ultimate driver—heat stress (not time)—we evaluated BSI-trait relationships at specific DHW levels instead of at survey time points. For each DHW level, we fit the linear regression between the instantaneous BSI (response variable) of each colony against their corresponding colony-level trait of interest (predictor variable). This allowed us to evaluate how the trait relationships change throughout the duration of the heat stress exposure, shown as changes in BSI-trait regression slopes. All LMs were fit using Bayesian inference in R-INLA[61], and uncertainty in regression slope estimations were quantified using Bayesian credible intervals (95%, 75%, 50%). These analyses can be reproduced using the open access code and data generated in this project.

**Inclusion and ethics statement**. The research presented here adhered to the ethical and inclusivity standards consistent with the corresponding author's institutional and internal review board policies.

**Reporting summary**. Further information on research design is available in the Nature Portfolio Reporting Summary linked to this article.

## Data availability

All original data and R code produced in this study are publicly available on Figshare at https://doi.org/10.25405/data.ncl.20411589. ITS2 sequences have been archived publicly at NCBI under BioProject 864615 (http://www.ncbi.nlm.nih.gov/bioproject/864615) and processed symbiont community composition can be explored publicly at https://symportal.org. Any additional information required to reanalyse the data reported in this paper is available from the lead contact upon request.

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

## Acknowledgements

We thank the numerous staff at the Palau International Coral Reef Centre (PICRC) who supported this research, Dr. Jamie Craggs for contributions to the experimental aquaria system, and Faith Paysinger for work on fecundity measurement. This research was funded by the Natural Environment Research Council's ONE Planet Doctoral Training Partnership (NE/S007512/1) to L.L. and the European Research Council Horizon 2020 project CORALASSIST (725848) awarded to J.R.G. The Reef Restoration and Adaptation Program is funded by the partnership between the Australian Government's Reef Trust and the Great Barrier Reef Foundation, which funded the time of R.F. on this project.

## Author contributions

The first draft of the manuscript, 3D model comparisons, and statistical analyses were conducted by L.L., with supervision from J.R.G., P.J.M., J.C.B., and H.K.E. The marine heatwave emulation experiment was conducted by A.H., H.M., J.B., A.J.E., E.S., and J.R.G. 3D photogrammetry surveys were conducted by R.F., W.F.F., D.R.P., B.S., and J.R.G., and photogrammetry models were constructed by D.R.P., R.F., W.F.F., and B.S. Symbiont DNA extractions were conducted by J.C.B. and A.H. Y.G. supported fieldwork. L.L., A.H., D.R.P., J.C.B., P.J.M., R.F., W.F.F., E.B., H.K.E., A.J.E., Y.G., H.M.M., B.S., E.S., and J.R.G. contributed to the development of ideas and writing the final manuscript.

## Competing interests

The authors declare no competing interests.
