## [Peer Review File · Communications Biology]

Reviewers' comments:

Reviewer #1 (Remarks to the Author):

Lachs et al. conducted a 5-week heatwave emulation experiment to assess coral thermal response using fragments from colonies of *Acropora digitifera*, sourced from a single reef in Palau. Experimental results were compared to growth, fecundity, and symbiont community composition of the same colonies monitored in situ. Overall, the authors did not find a trade-off between heat tolerance and growth or fecundity, and interestingly found a positive association between heat tolerance and growth.

The manuscript is well written and presents findings that will be of interest to coral biologists and reef managers alike. Overall, in my opinion the manuscript needs very little revision in order to be suitable for publication. Minor revisions to incorporate more detail in the methods would improve replicability and subtle clarification of the study design in the abstract would strengthen the manuscript. My detailed comments are given below; I wish the authors all the best with their revision.

Abstract

22-23: Slight rephrasing of this sentence is needed to describe the two components of this study more clearly, i.e., the experiment vs. the in situ assessment of growth, fecundity, and symbiont community. As it is currently written it reads as though all response metrics were assessed in the lab experiment.

Introduction & Discussion

No specific comments for these sections; overall the text is clear and provides good background for the topics discussed in this manuscript, as well as relevant interpretation of results based on the existing literature.

Results

119: typo; change 'vales' to 'values'

202-203: Fig 2 caption, should 'instantaneous bleaching mortality' be changed to 'bleaching survival index'?

Methods

While I do appreciate the clear overview of the experiment in Table S1, I think a few details should be added into the main text as often readers will not seek out information in the supplementary materials.

348: Please add the sample size (i.e., number of colonies) used in the experiment in this sentence.

352: What were the pre-experiment acclimation conditions (light, temperature)? The same as the experimental tanks?

352: Please indicate approximate ramping rate for heating, and state the type of heaters/temperature controllers used.

357: Please state the PAR

378: How many colonies died? Please indicate this clearly in the text for context.

390: Is the 0.75 cut-off defined by the authors? If yes, the current text explanation is clear and doesn't need any modification. If no, please add a citation for this defined cut-off.

413: Since not all colonies were imaged at each of the 3 time points (which is logistically understandable) this should likely be stated clearly in the text; just in brackets perhaps after the '114 3D constructions for 45 colonies ($n = x$ for all time points, and $n = x$ for 2 time points)'

Statistical analyses

I do not have a strong background in Bayesian statistics so for this component of the manuscript I will defer to other reviewer/editor discretion.

Figures are clear and well designed, and supplementary materials provide robust supporting

information for this experiment.

Reviewer #2 (Remarks to the Author):

This study investigates trade-offs and co-benefits across different traits including heat tolerance in a single coral species, *Acropora digitifera* using field measurements and aquaria experiments. This is an interesting study for understanding how a key reef-building coral species may persist under future ocean warming. However, there are some very critical considerations relating to growth, particularly the limitations of the approach used to estimate coral growth in this study, that absolutely must be addressed before this work can be published.

Growth measurements using photogrammetry reflects the change in the colony size in space and time but does not account for skeletal density or secondary skeletal infilling and therefore is not a measure of growth in terms of skeletal calcium carbonate accretion. For example, multiple corals could be growing at the same rate in terms of changes in colony size (surface area, volume), but some may be growing more dense skeletons than others, which is an important trade-off. Metrics of volume and surface area from 3D photogrammetry should not be confused as a measure of skeletal biomineralization (i.e. coral calcification), especially for branching acroporid corals that can have substantial secondary infilling and variation in skeletal density. This is a really important distinction and consideration given that this study aims to compare trade-offs between coral growth rates and heat tolerance. It is possible to quantify total net coral growth (calcification), for example using the buoyant weight technique (Jokiel 1978), which is typically normalised to surface area to allow comparisons between different locations, species and studies etc. Previous pivotal/keystone work on this topic used the buoyant weight technique, so it is possible that the contrasting results found here are due fundamental differences in the approach used to estimate growth. This must be acknowledged and properly accounted for throughout the manuscript, especially when comparing the results of this study with findings from previous work where totally different methods were used and they produce fundamentally different metrics of coral growth.

Line 81: A little confused by the wording of this sentence saying that the growth disadvantage is eliminated under extreme heating. Wouldn't the growth disadvantage be exacerbated?

Line 244: Need references here.

Line 248: Is this the first study to find no links between heat tolerance and growth?

Line 250: It was very weak positive relationship, though.

Line 254: There are some huge differences in the way that growth rates were measured in this study compared to the previous work that is being compared to here by ref 19 Jones and Berkelmans (2010). They used the buoyant weight technique (Jokiel 1978), which as stated above provides a more comprehensive measure of coral growth especially for understanding trade-offs in the energetic cost of biomineralization. The limitations of the study need to be mentioned up front and throughout, especially when comparing to other studies that use a completely different methodology for quantifying coral growth.

Line 271: Again, volume and surface area using 3D photogrammetry is not the same as coral calcification (in terms of mg CaCO₃ growth as per Marshall & Clode 2004) so the results may differ from Marshall & Clode (2004) and others with respect to thermal optima for growth.

Line 308: There is also the issue of a decreased winter reprieve and the consequences for coral health in terms of growth, disease, etc.

Line 357: How were the Hobo loggers calibrated? Include the precision.

Line 378: How was the health status scored visually? Was a colour chart used or something similar?

Figure 2: How are there negative values for growth in Figure 2? Also need to include letters to denote each part of the figure i.e. the different panels.

Reviewer #3 (Remarks to the Author):

This paper is nicely thought out and well-written and I appreciate the effort the authors put toward including examples from other ecosystems to set the stage for the study. The use of a range of corals with limited symbiont diversity is a strong system and does a good job isolating host-effects. This approach to evaluating tradeoffs will make a valuable contribution to the literature and our understanding of how coral adaptation is likely to impact coral reef function. The description and thoroughness of the methods (particularly for photogrammetry work) is excellent.

Overall, my lack of knowledge on Bayesian statistics hurt the interpretability of this paper and will likely be a sticking point for other readers as well, so it may limit the utility of this review. This choice, which I acknowledge is the authors' to make, is amplified by the very small effect sizes in their study. I came away from this paper thinking that the authors did a good job documenting the lack of tradeoffs between heat tolerance, growth and fecundity, but not that they convincingly demonstrated co-benefits, which is the cr. I hope a few methodological and statistical comments below might be helpful for resolving this.

Major points:

1. After looking at figure 1b I am concerned about the difference between blue/purple tanks and red/orange for downstream analysis and if the cause of the differences could be identified (light being my concern). The authors say elsewhere in the methods that they focus on DHW accumulated, but I am wondering if this was done on a tank-by-tank basis or averaged across a treatment? For example, by rough estimation it looks like these two tank groups hit 4DHW almost a week apart (maybe 3 weeks vs 3.75). Over the scale of this experiment that seems like an important shift and I think a justification of the approach used is warranted. This is introducing a bit of confusion for me because (for example) figure 3c shows time on the x-axis rather than accumulated DHW. I also note that the average BSI was used for the correlations later – I think this is a good enough method but if heat stress accumulation was lower in some tanks it may play a role.
2. I am curious what the authors think about the few extreme negative outliers in the growth analysis. While I assume that the removal of a few of these points doesn't impact the interpreted outcome (correct?), it's also pretty unclear to me that these example colonies are actually functioning biologically as part of the population in the way the authors intended – a difference in kind rather than degree. For example, in figure 2a if live surface area growth/year typically ranges from 0 to say 1000, what does a value of -1500 or -2000 even mean biologically? Figure S7 shows that these values are due to partial mortality and in Line 301 the authors point out that this mortality is likely related to other causes, which may represent more of a multi-stressor situation than a tradeoff situation. Overall, I think the inclusion of these points is problematic.
3. The statistical approach used here is complex. I acknowledge that my lack of understanding of Bayesian statistics hurts me here, but I would guess many readers will feel the same way. I have no objections if the reviewers and editors feel it is rigorous, but I wanted to point out that it hurts the comparability and interpretability of the results.
4. More directly, I have some concern that this approach is complicating what are effectively a range of null results, or at least very small effect sizes. Which would still be important evidence for the lack of tradeoffs, if not the co-benefits. As such, I think the title may be overselling the outcomes here a bit. For example, L187 "Contrary to expectations, we found positive associations between heat tolerance and growth metrics". The authors provide caveats in the next sentence, but if I am reading this correctly the slope of this relationship (e.g., Figure 2a bottom) is between -0.000012 and 0.000061. Perhaps this is just semantics (and I do see Figure S8) or my misunderstanding of scaling in the regression, but does this justify identifying co-benefits?
5. Figure 3a is a potentially important point regardless of the stats as 1 DHW differences is ecologically meaningful – I really think these data should be shown, not via schematic. Especially as it reinforces

the positive relationship overall. However, again, the negative growth colonies are also those with substantial partial mortality, which the authors may decide to address.

6. I am really unsure about the entirety of the analysis in Figure 3c from a practical and conceptual standpoint. Is there a reason to expect the relationship between these variables to change during increasing heat stress that is not captured by a more integrative metric that shows the onset at some amount of heat stress (i.e., BSI)? I hear the point about the two peaks representing bleaching onset and mortality onset, I just can't figure out if/how this particular analysis is meant to strengthen the relationships in figure 2?

Minor Notes:

P75: There are a couple of potentially important citations that I'd argue should be included here^{1, 2}. I also think acknowledgement and discussion of this paper³ is very important.

L 84: despite the results of the cited paper, I think this is broadly incorrect^{4, 5, 6}. I only point this out because the authors have chosen *Montipora* as a contrast point – other genera may offer this but I don't think this example stands.

L91: generally speaking I am not a fan of declaring this – I agree and can't come up with an example to contest, but I think framing it as a very important source of variation even if nested within genera-level symbiont community differences is enough

F1a – the key for MMM and MMM+1 is not legible (although obviously the solid line is the +1)

F1d – it would be good to have sample sizes by profile in this figure somewhere

L129:

Might clarify here, I think this critical point is the first observation at or below 0.75? Also might be worth citing experimental DHW for cross-referencing and standardization⁷

L153: I understand this is what the stats say, but the green profile is behaving quite different from an ecological perspective. Supplemental figures suggest this is a profile with very few samples, which might be worth pointing out in the main text.

References

1. Bay RA, Palumbi SR. Transcriptome predictors of coral survival and growth in a highly variable environment. *Ecology and Evolution*, 1-10 (2017).
2. Cornwell B, et al. Widespread variation in heat tolerance and symbiont load are associated with growth tradeoffs in the coral *Acropora hyacinthus* in Palau. *Elife* 10, e64790 (2021).
3. Wright RM, Mera H, Kenkel CD, Nayfa M, Bay LK, Matz MV. Positive genetic associations among fitness traits support evolvability of a reef-building coral under multiple stressors. *Global Change Biology* 25, 3294-3304 (2019).
4. Roach TN, Dilworth J, Jones AD, Quinn RA, Drury C. Metabolomic signatures of coral bleaching history. *Nature Ecology & Evolution*, 1-9 (2021).
5. Drury C, et al. Intrapopulation adaptive variance supports thermal tolerance in a reef-building coral.

Communications Biology 5, 1-10 (2022).

6. Drury C, Dilworth J, Majerová E, Caruso C, Greer JB. Expression plasticity regulates intraspecific variation in the acclimatization potential of a reef-building coral [dataset]. Zenodo DOI:10.5281/zenodo.6877825 (2022).

7. Leggat W, Heron SF, Fordyce A, Suggett DJ, Ainsworth TD. Experiment Degree Heating Week (eDHW) as a novel metric to reconcile and validate past and future global coral bleaching studies. Journal of Environmental Management 301, 113919 (2022).

From Editor

Dear Mr Lachs,

Your manuscript entitled "Co-benefits not trade-offs associated with heat tolerance in a reef building coral" has now been seen by 3 referees, whose comments are appended below. You will see from their comments copied below that while they find your work of potential interest, they have raised quite substantial concerns that must be addressed. In light of these comments, we cannot accept the manuscript for publication, but would be interested in considering a revised version that addresses these serious concerns.

We thank you and the reviewers for your interest in this piece of work, and for helpful and constructive comments. In this response to reviewers document, please note that all author responses are shown in **blue**. Quotes from the manuscript are shown in *italics*. Line numbers are shown for the simple view (tracked changes not showing). Revisions/additions compared to the last draft are shown as *underlined italics*.

We hope you will find the referees' comments useful as you decide how to proceed. Should further experimental data or analysis allow you to address these criticisms, we would be happy to look at a substantially revised manuscript. However, please bear in mind that we will be reluctant to approach the referees again in the absence of major revisions.

The referees have major concerns about the validity of photogrammetry to measure growth, due to the inability to account for secondary infilling and other possibilities. This will require some further work to address. Please note that none of the reviewers were comfortable reviewing the Bayesian statistical aspect. We may seek the opinion of a new reviewer in the next round to examine this aspect, but do not wish to delay the decision further at this time. We also advise presenting this aspect more clearly for non-expert readership.

In this response we discuss the different ways of measuring growth in corals (area/volume/calcification) and explain why photogrammetry was the most appropriate for this study. We have also included additional text in the manuscript to describe our statistical approach more clearly and to explain how our results relate to other studies that measure growth using skeletal density estimations from buoyant weight measurements.

We are committed to providing a fair and constructive peer-review process. Do not hesitate to contact us if you wish to discuss the revision or if there are specific requests from the reviewers that you believe are technically impossible or unlikely to yield a meaningful outcome.

If you decide to submit a revised version, we ask that you ensure your manuscript complies with our editorial policies. Please see our revision checklist for guidance on formatting the manuscript and complying with our policies. A comprehensive guide to our formatting requirements for final submissions is also available for your reference here.

We expect major revisions of this nature to take around six months to complete, but appreciate that every situation is unique. Please take as long as necessary to address these concerns in full, including performing any additional experimental work required. We look forward to receiving your revised manuscript when it is ready and will not enforce any specific deadline. However, please bear in mind that if the revision process takes significantly longer than six months, we will need to confirm that nothing similar has been accepted for publication at Communications Biology or published elsewhere in the meantime.

Please do not hesitate to contact me if you have any questions or would like to discuss the required revisions further. Thank you for the opportunity to review your work.

Best regards,

Luke R. Grinham, PhD

We thank you and the reviewers for your interest in this piece of work, and for helpful and constructive comments. Addressing these points has certainly improved the manuscript and we hope that you find our changes, clarifications, and additions suitable for publication. Please find all point-by-point responses and revisions below.

Reviewers' comments:

Reviewer #1 (Remarks to the Author):

Lachs et al. conducted a 5-week heatwave emulation experiment to assess coral thermal response using fragments from colonies of *Acropora digitifera*, sourced from a single reef in Palau. Experimental results were compared to growth, fecundity, and symbiont community composition of the same colonies monitored in situ. Overall, the authors did not find a trade-off between heat tolerance and growth or fecundity, and interestingly found a positive association between heat tolerance and growth.

The manuscript is well written and presents findings that will be of interest to coral biologists and reef managers alike. Overall, in my opinion the manuscript needs very little revision in order to be suitable for publication. Minor revisions to incorporate more detail in the methods would improve replicability and subtle clarification of the study design in the abstract would strengthen the manuscript. My detailed comments are given below; I wish the authors all the best with their revision.

Abstract

22-23: Slight rephrasing of this sentence is needed to describe the two components of this study more clearly, i.e., the experiment vs. the in situ assessment of growth, fecundity, and symbiont community. As it is currently written it reads as though all response metrics were assessed in the lab experiment.

Response: This is a great point. We have made relevant changes to the abstract.

Lines 23: *We tested for trade-offs between heat tolerance and three traits measured from the colonies in situ...*

Introduction & Discussion

No specific comments for these sections; overall the text is clear and provides good background for the topics discussed in this manuscript, as well as relevant interpretation of results based on the existing literature.

Results

119: typo; change 'vales' to 'values'

Response: Thanks for spotting this. The relevant change has been made on **Line 123**.

202-203: Fig 2 caption, should 'instantaneous bleaching mortality' be changed to 'bleaching survival index'?

Response: Thanks for this suggestion. Please note that the text on this line is actually referring to Figure 3, not Figure 2. As suggested, we have now clarified that we are referring to the bleaching and survival responses throughout the heat stress exposure, measured as BSI.

Lines 212-213: ... *instantaneous bleaching and survival responses (measured as BSI) were variable...*

Methods

While I do appreciate the clear overview of the experiment in Table S1, I think a few details should be added into the main text as often readers will not seek out information in the supplementary materials.

348: Please add the sample size (i.e., number of colonies) used in the experiment in this sentence.

Response: Thanks for highlighting this point. We have made the suggested changes to text at the relevant points in the following methodological subsection, as the suggested section entitled "4.2 / Marine heatwave emulation experiment" we only describe the environmental conditions, not information on the biological samples.

Lines 410-411: *In total, fragments from 66 of 70 colonies were exposed to the assay, as four colonies were not found during collection.*

352: What were the pre-experiment acclimation conditions (light, temperature)? The same as the experimental tanks?

Response: We have clarified that the thermal conditions were ambient with no heating across all tanks. We have also clarified that lights were used both in the acclimation period and the stress exposure period.

Lines 379: ... *7–10-day acclimatisation period under ambient thermal conditions in all tanks.*

Lines 385-386: ... *light conditions during the acclimatisation period and the heat stress exposure ...*

352: Please indicate approximate ramping rate for heating, and state the type of heaters/temperature controllers used.

Response: We have addressed this by adding the relevant details, showing the approximate heating rate, and guiding readers to the supplementary materials for more detailed information. The experimental tank setup followed the same design explained in our recent paper in Proceedings B (Humanes & Lachs *et al.* 2021). As such, we have referred to this study where the entire setup is explained in much more detail. This improves the clarity of our experiment whilst also saving on space. Readers interested in more detail should be able to find it easily by following these links.

Lines 380-383: ... *temperature was increased gradually (approximately 0.8 °C/week, Table S1) over the time course of the experiment (35 days), reaching a final bleaching-level temperature of approximately 33 °C, or 3.5 °C above the local climatological baseline (MMM – maximum of monthly means, Fig. 1A, Table S1, see ⁸ for more detailed tank setup description.*

357: Please state the PAR

Response: The information has been added in this sentence.

Lines 385-387: *while aquarium lights provided light conditions during the acclimatisation period and the heat stress exposure at a daily average intensity of $400 \mu\text{mol m}^{-2} \text{s}^{-1}$, corresponding to the average light intensity measured in Mascherchur at midday...*

378: How many colonies died? Please indicate this clearly in the text for context.

Response: The information has been added in this sentence.

Line 415: *... was not assigned a heat tolerance score (2 colonies).*

390: Is the 0.75 cut-off defined by the authors? If yes, the current text explanation is clear and doesn't need any modification. If no, please add a citation for this defined cut-off.

Response: This cut-off is not something used by other authors and has been defined here for the purposes of this study. We rephrased this sentence to with a citation to clarify this point.

Lines 429-431: *This BSI score was chosen as it represents a colony with an average health status of 'partially bleached' across all replicate fragments, a health status indicating that bleaching and mortality responses have started and will progress further⁸.*

413: Since not all colonies were imaged at each of the 3 time points (which is logistically understandable) this should likely be stated clearly in the text; just in brackets perhaps after the '114 3D constructions for 45 colonies (n = x for all time points, and n = x for 2 time points)'

Response: The information has been added in this sentence.

Lines 460: *45 coral colonies (n = 24 for all time points, and n = 21 for two time points).*

Statistical analyses

I do not have a strong background in Bayesian statistics so for this component of the manuscript I will defer to other reviewer/editor discretion.

Figures are clear and well designed, and supplementary materials provide robust supporting information for this experiment.

Response: Thanks for the great comments that have improved the manuscript. We hope that our changes in relation to the statistics have made the analysis easier for readers that are not familiarised with Bayesian methods easier to follow.

Reviewer #2 (Remarks to the Author):

This study investigates trade-offs and co-benefits across different traits including heat tolerance in a single coral species, *Acropora digitifera* using field measurements and aquaria experiments. This is an interesting study for understanding how a key reef-building coral species may persist under future ocean warming. However, there are some very critical considerations relating to growth, particularly the limitations of the approach used to estimate coral growth in this study, that absolutely must be addressed before this work can be published.

Growth measurements using photogrammetry reflects the change in the colony size in space and time but does not account for skeletal density or secondary skeletal infilling and therefore is not a measure of growth in terms of skeletal calcium carbonate accretion. For example, multiple corals could be growing at the same rate in terms of changes in colony size (surface area, volume), but

some may be growing more dense skeletons than others, which is an important trade-off. Metrics of volume and surface area from 3D photogrammetry should not be confused as a measure of skeletal biomineralization (i.e. coral calcification), especially for branching acroporid corals that can have substantial secondary infilling and variation in skeletal density.

Response: We thank the reviewer for picking up on this possibility of alternative trade-offs with traits not measured in this study (e.g., skeletal density), and the distinction between the metric of growth used in our study and calcification rates. To address this comment, we added text that describes possible alternative trade-offs and we also highlight the distinction between the measure of whole colony growth we have used in this study compared to other studies that measure growth in terms of calcification rates. We have added both to the main intro, discussion, and methods.

Lines 85-88: Technological advances in photogrammetry now allow completely non-invasive determination of colony growth, which has advantages for repeated monitoring of corals in the field compared other techniques (e.g., buoyant weight) which require removing corals from the substrate or causing potential harm ²⁷.

Lines 329-333: It may be possible that while we find weak positive associations between heat tolerance and whole colony growth, trade-offs with other traits such as calcification could still exist ¹⁶. Such a trade-off could compromise individual fitness of more heat tolerant corals particularly during storm surges when there is a higher risk of colony breakage.

Lines 448-457: Previous studies investigating growth-heat tolerance trade-offs in corals have measured growth as calcification rates based on Calcium incorporation ⁴⁰ or buoyant weight techniques ¹⁶, which are both influenced by skeletal density and secondary skeletal infilling. These methods are invasive and require manipulation of the coral colony, which can potentially influence coral fitness ²⁷. Moreover, as colonial organisms, it is possible for corals to experience shrinkage and still survive (i.e., partial mortality or reduction in size) and shift between net positive or negative growth over multiple occasions. This phenomenon may be undetected using some growth measurement techniques (e.g., buoyant weight). Therefore, at the colony level we deemed it more appropriate to use photogrammetry, a non-invasive method of measuring growth that can capture any changes in colony size with high accuracy and precision ^{57,58}.

This is a really important distinction and consideration given that this study aims to compare trade-offs between coral growth rates and heat tolerance. It is possible to quantify total net coral growth (calcification), for example using the buoyant weight technique (Jokiel 1978), which is typically normalised to surface area to allow comparisons between different locations, species and studies etc. Previous pivotal/keystone work on this topic used the buoyant weight technique, so it is possible that the contrasting results found here are due fundamental differences in the approach used to estimate growth. This must be acknowledged and properly accounted for throughout the manuscript, especially when comparing the results of this study with findings from previous work where totally different methods were used and they produce fundamentally different metrics of coral growth.

Response: Thanks for highlighting this point. We agree that there might be a trade-off between heat tolerance and calcification/skeletal density that we are not picking up with our methodological approach. We would firstly like to clarify the reasoning behind our choice of using photogrammetry to measure growth. Because we aimed to keep these colonies on the reef (without detaching them) for a multi-year period, it would have been impossible to measure calcification rates (including infilling) for these whole colonies using buoyant weight techniques, as that would have required

removing the corals from the substrate. Instead, we chose to use a non-invasive method of characterising whole colony growth using photogrammetry. Notably, there are different links to fitness with the different measures of growth that one could use. High growth in terms of calcification rate may improve fitness by enhanced survivorship against storm disturbance, while high growth in terms of surface area change may improve fitness based on reproductive output as larger colonies produce more eggs. It might also increase the speed with which corals reach a size escape from some forms of corallivory. Importantly, as corals are colonial organisms, it is possible for corals to experience shrinkage (i.e., partial mortality or negative growth) and still survive, subsequently returning to positive growth, potentially over multiple occasions. Such negative growth may be undetected using some measurement techniques. For instance, repeated buoyant weight measurements of a colony may not account for loss of live tissue over some of the colony (while skeleton remains intact), whereas photogrammetry methods where dead tissue areas can be removed from digital 3D models post hoc would be able to account for this. Therefore, at the colony level we deemed it more appropriate to use a method of measuring growth that can capture negative growth, i.e., photogrammetry.

We have addressed these comments by adding text on the distinction between different methods used to measure coral colony growth and providing the caveat to our comparison with previous studies of trade-offs that used these other methods. See comments above (**Lines 448-457**), and the following.

Lines 270-274: *Previous work has identified considerable trade-offs between heat tolerance and growth in terms of calcification rates caused by the presence of different symbionts¹⁶, flagging this as a potential barrier to successful coral adaptation under climate change¹⁹. Our results show that heat tolerance and whole colony growth (in terms of surface area and volume) can be positively associated ...*

Lines 309-311: *However, our results cannot be compared directly to calcification-based studies since we have measured growth as changes in colony size (to capture net positive and negative changes) which may bear different implications for coral populations.*

Line 81: A little confused by the wording of this sentence saying that the growth disadvantage is eliminated under extreme heating. Wouldn't the growth disadvantage be exacerbated?

Response: The study by Cuning et al (2015) showed that the growth rates of clade C corals reduce to the level of clade D corals under increasing temperature treatments (Figure 2 in their paper). We have rephrased the sentence to make this clearer.

Lines 78-81: *Notably, this growth disadvantage can be eliminated under warming of 1.5–3 °C, as growth rates decline disproportionately with increasing temperature for corals hosting Cladocopium symbionts compared to those hosting Durusdinium symbionts²³.*

Line 244: Need references here.

Response: We have added relevant references here.

Line 248: Is this the first study to find no links between heat tolerance and growth?

Response: There is a recent study by Wright who have found similar positive trait correlations. As such we reference them and mention their study.

Lines 70-72: Such positive phenotypic correlations can also be associated to genetic correlations among traits which manifest as co-tolerance of individual organisms to multiple biotic and abiotic stressors¹⁵.

Lines 274-276: This builds on the recent finding of co-tolerance of individual corals to multiple stressors (e.g., thermal stress, bacterial infection)¹⁵.

Line 250: It was very weak positive relationship, though.

Response: We highlight this by adding to the text.

Lines 267: ... we found weak positive associations ...

Line 254: There are some huge differences in the way that growth rates were measured in this study compared to the previous work that is being compared to here by ref 19 Jones and Berkelmans (2010). They used the buoyant weight technique (Jokiel 1978), which as stated above provides a more comprehensive measure of coral growth especially for understanding trade-offs in the energetic cost of biomineralization. The limitations of the study need to be mentioned up front and throughout, especially when comparing to other studies that use a completely different methodology for quantifying coral growth.

Response: We have addressed this point in the following places, in some cases by referring to “colony” growth (i.e. the whole colony change in surface area or volume) to make this distinct from generic growth that could be mistaken for calcification. We have also included additional text describing the possible trade-off with calcification rates.

Lines 77-78: Ultimately this results in reduced coral growth in terms of calcification rates.

Lines 267: weak positive associations between heat tolerance and colony growth.

Lines 271: ... growth in terms of calcification rates caused ...

Lines 273-274: our results show that heat tolerance and whole colony growth (in terms of surface area and volume) can be positively associated.

Lines 298-299: confer higher tolerance at the expense of photosynthetic energetics and ultimately growth as calcification.

Lines 305-306: either being resistant to high temperatures or showing enhanced calcification rates.

Lines 325: heat tolerance and colony growth remained

Lines 329-333: It may be possible that while we find weak positive associations between heat tolerance and whole colony growth, trade-offs with other traits such as calcification could still exist¹⁶. Such a trade-off could compromise individual fitness of more heat tolerant corals particularly during storm surges when there is a higher risk of colony breakage.

Line 271: Again, volume and surface area using 3D photogrammetry is not the same as coral calcification (in terms of mg CaCO₃ growth as per Marshall & Clode 2004) so the results may differ from Marshall & Clode (2004) and others with respect to thermal optima for growth.

Response: Thanks for highlighting the distinction between the different forms of growth (surface area / volume / calcification). We have added to the text to clarify that we are discussing growth in terms of colony size (surface area or volume). Note that we have left the sentence about thermal

optima unchanged as this already specifies that the research on thermal optima has focused on calcification and photosynthesis.

Lines 283-286: *In general, we found that more heat tolerant individuals also tended to have higher colony growth rates. This implies that post-bleaching coral populations may not necessarily have lower overall growth in terms of colony size.*

Line 308: There is also the issue of a decreased winter reprieve and the consequences for coral health in terms of growth, disease, etc.

Response: This is a good point. As our study measured coral growth over the whole annual cycle, any winter reprieve should be accounted for. We already refer to this winter reprieve with the text “heat tolerance is unlikely to directly benefit corals during cooler months”. Notably we do not refer to winter, as for this tropical location there is a more dry-wet season annual cycle rather than summer/winter.

Line 357: How were the Hobo loggers calibrated? Include the precision.

Response: The relevant text has been added to the manuscript.

Lines 387-389: HOBO loggers, with 0.14 °C resolution and 0.45 °C accuracy, were calibrated against a RBR TR-1050 using the average offset for temperatures between 27 and 35 °C in increments of 0.5 °C.

Line 378: How was the health status scored visually? Was a colour chart used or something similar?

Response: Note this was not done using a colour chart. Pale corals still with some pigment were recorded as healthy, while bleaching was only considered when there was stark white with live tissue. We added text to describe this and highlight the previous work done on this method by correlating the health status scores with pigment concentrations and symbiont densities.

Lines 415-418: The health status of each fragment was scored visually into five categories based on stark whiteness and tissue state (see below) at intervals of between 1 and 3 days (total of 16 timepoints over 35 days). Notably the bleaching scores were highly correlated to pigment concentration and symbiont density⁸.

Figure 2: How are there negative values for growth in Figure 2? Also need to include letters to denote each part of the figure i.e. the different panels.

Response: As a colonial organism, shrinkage (negative growth) is possible for whole coral colonies and can through loss of live tissue and/or skeleton, which is accounted for in measurement of surface area and volumetric growth. We have described this further in the main text and methods to explain this. We have already labelled the figure using letters A and B for the left and right columns, and as such have left it this way.

Lines 202-204: These results were markedly similar even when colonies that had experienced shrinkage (reduction in surface area or volume) were excluded from the analysis (Fig. S10).

Lines 323-327: Yet even when colonies that had experienced shrinkage (reduction in surface area or volume) were removed from the analysis a weak positive association between heat tolerance and colony growth remained, suggesting the trend observed in this study was not an artefact of colonies undergoing shrinkage (e.g., through processes including predation, tissue necrosis, or breakage).

Lines 448-457: Previous studies investigating growth-heat tolerance trade-offs in corals have measured growth as calcification rates based on Calcium incorporation⁴⁰ or buoyant weight

techniques ¹⁶, which are both influenced by skeletal density and secondary skeletal infilling. These methods are invasive and require manipulation of the coral colony, which can potentially influence coral fitness ²⁷. Moreover, as colonial organisms, it is possible for corals to experience shrinkage and still survive (i.e., partial mortality or reduction in size) and shift between net positive or negative growth over multiple occasions. This phenomenon may be undetected using some growth measurement techniques (e.g., buoyant weight). Therefore, at the colony level we deemed it more appropriate to use photogrammetry, a non-invasive method of measuring growth that can capture any changes in colony size with high accuracy and precision ^{57,58}.

Reviewer #3 (Remarks to the Author):

This paper is nicely thought out and well-written and I appreciate the effort the authors put toward including examples from other ecosystems to set the stage for the study. The use of a range of corals with limited symbiont diversity is a strong system and does a good job isolating host-effects. This approach to evaluating tradeoffs will make a valuable contribution to the literature and our understanding of how coral adaptation is likely to impact coral reef function. The description and thoroughness of the methods (particularly for photogrammetry work) is excellent.

Overall, my lack of knowledge on Bayesian statistics hurt the interpretability of this paper and will likely be a sticking point for other readers as well, so it may limit the utility of this review. This choice, which I acknowledge is the authors' to make, is amplified by the very small effect sizes in their study. I came away from this paper thinking that the authors did a good job documenting the lack of tradeoffs between heat tolerance, growth and fecundity, but not that they convincingly demonstrated co-benefits, which is the cr. I hope a few methodological and statistical comments below might be helpful for resolving this.

Response: Bayesian approaches to fitting linear regressions allow for the inspection of posterior distributions for the estimated parameters (e.g., intercept or slope) and calculation of for example 95% credible intervals. This allows us to quantify the uncertainty in a more intuitive way than would have been possible using frequentist alternatives. For example, by analysing the posterior distribution of slope estimate (bell curve) we can say "there is a xx % chance that the value of the slope is negative". We therefore stayed with the Bayesian approach in the revised manuscript and have added text to explain the methods and results in more detail, so they are more easily accessible to the broad readership of *Communications Biology*.

Major points:

1. After looking at figure 1b I am concerned about the difference between blue/purple tanks and red/orange for downstream analysis and if the cause of the differences could be identified (light being my concern). The authors say elsewhere in the methods that they focus on DHW accumulated, but I am wondering if this was done on a tank-by-tank basis or averaged across a treatment? For example, by rough estimation it looks like these two tank groups hit 4DHW almost a week apart (maybe 3 weeks vs 3.75). Over the scale of this experiment that seems like an important shift and I think a justification of the approach used is warranted. This is introducing a bit of confusion for me because (for example) figure 3c shows time on the x-axis rather than accumulated DHW. I also note that the average BSI was used for the correlations later – I think this is a good enough method but if heat stress accumulation was lower in some tanks it may play a role.

Response: Thanks for the comments on this point. We would like to clarify that DHW was calculated on a per tank basis (**Lines 391-392**). As you indicated, there was a lagged accumulation of DHW in certain tanks which may have led to a lag in the progression of health status scores for individual fragments. We already addressed this in our analysis by conducting an alignment of health status scores of each fragment to 21 fixed DHW points (rather than timepoints). Note that this was conducted prior to calculating BSI and average BSI, so the lagged DHW accumulation in certain tanks would not affect the BSI values used for analysis (Fig. 3) or the average BSI values (Fig. 2). Despite mentioning this in our methods, we see that it was not explained clearly enough (we just cited our previous paper that used the same method). As such, we have added further descriptions of this method using a citation to link to the more detailed description already published (**Lines 432-434**). Moreover, the fragments of each colony were spread evenly across tanks, therefore any biases

would also be equal across all colonies by calculating BSIs and average BSIs across fragments from these different tanks. In fact, BSIs can also be calculated for each timepoint, and this is what is shown in Figure 1c. We clarify this difference in the figure caption (**Lines 146-147**).

Lines 391-392: *we calculated heat stress for each tank using the Degree Heating Weeks (DHW) metric.*

Lines 432-434: *To remove potential biases relating to lagged DHW profiles among tanks, we followed the method outlined in full detail in ⁸ which aligns DHW profiles among tanks and interpolates health status scores at fixed DHW values with fixed intervals, providing unbiased BSI values among colonies.*

Lines 145-146: *(D) The BSI-DHW relationship (where BSIs are corrected for DHW drift among tanks) was unaffected by symbiont ITS2 type.*

2. I am curious what the authors think about the few extreme negative outliers in the growth analysis. While I assume that the removal of a few of these points doesn't impact the interpreted outcome (correct?), it's also pretty unclear to me that these example colonies are actually functioning biologically as part of the population in the way the authors intended – a difference in kind rather than degree. For example, in figure 2a if live surface area growth/year typically ranges from 0 to say 1000, what does a value of -1500 or -2000 even mean biologically? Figure S7 shows that these values are due to partial mortality and in Line 301 the authors point out that this mortality is likely related to other causes, which may represent more of a multi-stressor situation than a tradeoff situation. Overall, I think the inclusion of these points is problematic.

Response: Thanks for raising this concern. We want to emphasise that the measure of growth we focus on here is whole colony growth, compared to other studies on trade-offs that have assessed growth as buoyant weight or size of fragments. The change in live surface area of a colony is a biologically meaningful metric, as it is closely related to the number of polyps on the colony and in turn colony fecundity, and the colony's competitive ability (e.g., smaller corals more prone to overtopping by faster growing colonies or algae). In regard to the possibility of negative values of growth (e.g., change in live surface area of the colony), this can occur if some of the tissue is lost (for example from breakage or necrosis). However, this is distinct from survival, as many colonies that suffer from partial loss of tissue can later recover and shift back to positive growth again. To address your comment regarding negative outliers, we re-ran the trade-off analyses on only the colonies with positive growth values and found the same lack of a trade-off between heat tolerance and growth, with a weak positive association with a slightly higher level of uncertainty (84% chance of positive slope, compared to 94% chance based on all colonies). As such, we have included this additional analysis in the supplementary materials (**revised Fig. S10**) and have mentioned it in the manuscript. This does not change the overall story of this study.

Lines 202-204: *These results were markedly similar even when colonies that had experienced shrinkage (reduction in surface area or volume) were excluded from the analysis (Fig. S10).*

Lines 323-327: *Yet even when colonies that had experienced shrinkage (reduction in surface area or volume) were removed from the analysis a weak positive association between heat tolerance and colony growth remained, suggesting the trend observed in this study was not an artefact of colonies undergoing shrinkage (e.g., through processes including predation, tissue necrosis, or breakage).*

3. The statistical approach used here is complex. I acknowledge that my lack of understanding of Bayesian statistics hurts me here, but I would guess many readers will feel the same way. I have no

objections if the reviewers and editors feel it is rigorous, but I wanted to point out that it hurts the comparability and interpretability of the results.

Response: We appreciate that using Bayesian statistical methods over frequentist alternatives is only recently becoming more established, yet the packages we are using are proven methods (see Rue et al 2009 - Approximate Bayesian inference for latent Gaussian models by using integrated nested Laplace approximations). The specific models we are fitting in this study are basic linear regressions, without complicated error structures etc.

The frequentist approach to fitting linear regressions (e.g., effect of variable X on variable Y) provides a rather ambiguous treatment of so-called 'non-significant' results (particularly let us consider the regression slope, B_1). This simply means that the analysis failed to find an effect of X on Y, less than a critical P value (alpha, typically set at 0.05 from convention). Yet this does not necessarily mean there is 'no effect' of X on Y. To obtain real clarity on the level of evidence our data provides for the relationship between the two variables, we need to quantify the posterior density function of the slope value directly.

Bayesian statistics do this by providing the posterior distribution of parameter estimates. This then allows us to investigate the shape of this distribution, and for instance ask more nuanced questions, such as "what is the probability that there is no trade-off?" or in other words, "what is the probability that the slope value is greater than zero". This gives a much more intuitive understanding as to our certainty in the results. As such, we have stayed with the Bayesian statistical approach, and to enhance the clarity of this approach for readers not used to thinking about Bayesian statistics concepts, we have added some extra sentences to the introduction and methodology sections.

Lines 114-117: We employ Bayesian methods for solving simple trait trade-off linear regressions (in the form: $\text{heat tolerance} \sim B_0 + B_1 \times \text{trait} + \text{error}$) to allow an intuitive quantification of uncertainty via inspection of posterior distributions, specifically testing the odds of no trade-off occurring (i.e., B_1 slope value > 0).

Lines 520-525: Uncertainty around a model parameter (e.g., the regression slope) is commonly described from using the lower and upper 95% credible intervals, which bound 95% of the area under a posterior distribution density curve. To calculate the probability of no trade-off between heat tolerance and other traits (i.e., a flat line or even a positive association with a slope greater than zero), we measured the proportion of the posterior distribution which exceeds zero.

4. More directly, I have some concern that this approach is complicating what are effectively a range of null results, or at least very small effect sizes. Which would still be important evidence for the lack of tradeoffs, if not the co-benefits. As such, I think the title may be overselling the outcomes here a bit. For example, L187 "Contrary to expectations, we found positive associations between heat tolerance and growth metrics". The authors provide caveats in the next sentence, but if I am reading this correctly the slope of this relationship (e.g., Figure 2a bottom) is between -0.000012 and 0.000061. Perhaps this is just semantics (and I do see Figure S8) or my misunderstanding of scaling in the regression, but does this justify identifying co-benefits?

Response: Thanks for highlighting this point. Please note that the highly uncertain result referenced here (Fig 2) is based on the heat tolerance summary statistic (average BSI). This result of "marginal significance" led us to conduct the further analysis on direct measures of BSI (Fig 3, not just the average BSI which could lose variations in the shape of response trajectory throughout a heat stress event) in order to investigate if there is any substance to the Figure 2 result. The strongest evidence for co-benefits comes from the final analysis in figure 3, showing the delayed onset of bleaching in

positive-growth colonies with a very high level of certainty (95% credible intervals not intersecting zero at DHW of approx. 4 C-weeks). Thus, given the strong evidence provided in Fig 3, we retain the discussion points about co-benefits.

For the example that the reviewer has identified above (volumetric growth, fig2a bottom), a slope of $2.4e^{-5}$ translates to the y variable increasing by $2.4e^{-5}$ for a x variable increase of 1. Considering that growth values typically range from -500 to +1500 cm^3/yr (with extreme values outside this range), let us make a comparison of BSI for two hypothetical coral colonies with those two growth values. Based on our regression model we would predict that a difference in average BSI values between a colony that is reducing in size at $-500 \text{ cm}^3/\text{yr}$, versus a colony that is growing at $+1000 \text{ cm}^3/\text{yr}$ would equate to 0.04 BSI units ($1000 \times 2.4e^{-5} - (-500 \times 2.4e^{-5})$). Centred around the mean heat tolerance value (average BSI = 0.74), a difference of 0.04 BSI units (average BSI of 0.72 versus 0.76) translates to a difference between the 40th and 60th percentiles of the population. As such we consider this slope to be weak yet potentially important.

The purpose of Figure 2 is therefore to illustrate the need for a more detailed analysis of the heat stress response, which we subsequently provided in Figure 3. We have included more detail in the manuscript to clarify any ambiguities about the scale of the slopes in fig 2a, we have included more detail in the manuscript.

Lines 197-202: *Although the slope values seem small (i.e., $\times 10^{-5}$) this is due to a disparity in the order of magnitude between growth values ($\times 10^3$) and average BSI values ($\times 10^{-1}$). Accordingly, these differences in growth correspond to weak but potentially important shifts in average BMI. For instance, moving from the 10th to 90th percentiles of volumetric colony growth correspond to a shift in heat tolerance from the 40th to 60th percentile of the population (Fig. 2A), or an increased bleaching heat stress tolerance of 0.7–0.9 °C-weeks (Fig. S9).*

5. Figure 3a is a potentially important point regardless of the stats as 1 DHW differences is ecologically meaningful – I really think these data should be shown, not via schematic. Especially as it reinforces the positive relationship overall. However, again, the negative growth colonies are also those with substantial partial mortality, which the authors may decide to address.

Response: This is a great point. We tried different ways of showing this trend using the data, but it is quite a challenge, and we think that the figure 3 in this final draft is probably the best representation of the data. For instance, in the schematic we show how the onset of bleaching and mortality differ between positive growth and those colonies that experienced shrinkage (negative growth). We show the comparison between positive-growth and negative-growth colonies as if they are two completely distinct groups. However, in reality there is a continuous distribution of growth values. That is why we opted to assess how growth influences the bleaching survival response (BSI) as a series of regressions at different stages throughout the heat stress.

As such we have not changed figure 3 but we have added an additional figure which displays the bleaching dosage for each colony versus growth. Due to the arbitrary definition of the bleaching dosage at BSI = 0.75, we show a sensitivity analysis of this figure for multiple BSI cut-offs from 0.8 to 0.75. Notably for mortality, an equivalent metric could not be computed (i.e., mortality dosage), as many colonies do not reach mortality (Fig. S9).

Lines 214-216: *at a DHW value of approximately 4–6 °C-weeks (Fig. 3A, Fig. S9). However, bleaching onset was delayed in fragments of positive-growth colonies compared to negative-growth colonies by approximately 1 °C-week (Fig. 3A, Fig. S9).*

6. I am really unsure about the entirety of the analysis in Figure 3c from a practical and conceptual standpoint. Is there a reason to expect the relationship between these variables to change during increasing heat stress that is not captured by a more integrative metric that shows the onset at some amount of heat stress (i.e., BSI)? I hear the point about the two peaks representing bleaching onset and mortality onset, I just cant figure out if/how this particular analysis is meant to strengthen the relationships in figure 2?

Response: Thanks for the expansion of ideas on this point. Following your suggestion, we have calculated a bleaching dosage of heat stress for each colony based a number of different bleaching BSI cut offs (0.8 to 0.75). These results have now been added to the supplementary materials (**Fig. S9**) and have been referenced in the results as such. However, we note the limitation of using a “bleaching dosage” style of metric as it summarises the BSI values (similar to average BSI). The strength of the analysis shown in Figure 3 is that it is based on the raw BMI values (not summary statistics). While the result in figure two suggests a weak co-benefit, but with high uncertainty (credible intervals overlapping zero), the result in Fig 3c shows the nuance of this co-benefit. It is at the onset of bleaching and at the onset of mortality, that there is a maximum co-benefit (greatest slope value), and at the onset of bleaching (first peak) that there is a high level of certainty (credible intervals not overlapping zero). As such we can say that yes there is a weak co-benefit. For now, we have left the title as it is, however, we are happy to have further discussion with the editor and reviewers regarding whether to change it to “No apparent trade-offs associated with heat tolerance in a reef-building coral”. We are happy to have added Figure S9 that explains more clearly the delayed bleaching onset response as a function of growth as shown conceptually in Fig 3a.

Lines 247-252: *The strength of this temporal analysis (Fig. 3) is that it is based on raw BSI values (not summary statistics like average BSI). While the analysis of average BSI (Fig. 2), suggests a weak co-benefit between heat tolerance and growth but with high uncertainty, the temporal analysis (Fig. 3C) shows the nuance of this co-benefit. It is at the onset of bleaching and at the onset of mortality that there is a maximum co-benefit (greatest slope value), and at the onset of bleaching that there is the highest level of confidence of this co-benefit (95% credible intervals not overlapping zero).*

Minor Notes:

P75: There are a couple of potentially important citations that I’d argue should be included here^{1, 2}. I also think acknowledgement and discussion of this paper³ is very important.

Response: Thanks for linking us to these citations. We have added them and including the Wright et al study in in our introduction and discussion.

Lines 70-72: *Such positive phenotypic correlations can also be associated to genetic correlations among traits which manifest as co-tolerance of individual organisms to multiple biotic and abiotic stressors¹⁵.*

Lines 274-276: *This builds on the recent finding of co-tolerance of individual corals to multiple stressors (e.g., thermal stress, bacterial infection)¹⁵.*

L 84: despite the results of the cited paper, I think this is broadly incorrect^{4, 5, 6}. I only point this out because the authors have chosen Montipora as a contrast point – other genera may offer this but I don’t think this example stands.

Response: Thanks to the reviewer for these references. As such we have updated our statement to also reflect the work from Roach et al (25) on Montipora in addition to the work from Matthews et al (26).

Lines 83-85: *However, for other coral genera (e.g., Montipora sp.), there is mixed evidence on whether (see ²⁵) or not (see ²⁶) Durusdinium spp. symbionts (rather than Cladocopium spp.) influence coral host physiology and metabolism.*

L91: generally speaking I am not a fan of declaring this – I agree and can't come up with an example to contest, but I think framing it as a very important source of variation even if nested within genera-level symbiont community differences is enough

Response: We apologise, but we were unsure which specific part of the manuscript this comment relates to. Line 91 of the initial submission corresponds to this sentence: "However, it is yet to be tested whether trade-offs between heat tolerance and other ecological traits exist for corals that share the same Symbiodiniaceae community.". However, we do not think the comment directly addresses this text. Therefore, have not made any changes directly based on this comment but will gladly make further revisions if needed upon clarification.

F1a – the key for MMM and MMM+1 is not legible (although obviously the solid line is the +1)

Response: This has been amended see Fig 1a.

F1d – it would be good to have sample sizes by profile in this figure somewhere

Response: This has been amended see Fig 1d and caption.

Lines 146-147: *... showing number of colonies with each symbiont ITS2 type in brackets.*

L129: Might clarify here, I think this critical point is the first observation at or below 0.75? Also might be worth citing experimental DHW for cross-referencing and standardization⁷

Response: Yes you are correct, this refers to the first fixed DHW value (of the 21 different fixed levels of equal interval on which BMIs were interpolated) at which BMI \leq 0.75. We have clarified this in the text (**Lines 132-133**). Also, thanks for highlighting the Leggat study. Here we found that using the eDHW method (using the satellite MMM) actually underestimated heat stress in our tanks. Adjusting the MMM based on in situ data was needed to make our experimental DHWs comparable to the NOAA bleaching forecasts. We have addressed this in **Lines 401-404**. Notably, we also found that the text on lines 418 was relevant to the previous methodological section (MMM adjustment) and have moved it there (**Lines 404-406**).

Lines 132-133: *onset of bleaching and mortality (first fixed DHW value at which BSI \leq 0.75)*

Lines 401-404: *This builds upon the eDHW method which suggest using the satellite based MMM to compute experimental DHWs ⁴⁹. However, our previous work on Mascherchur reef has found that the eDHWs underestimate true DHWs due to a mismatch between the satellite data and in situ reef conditions ⁸.*

Lines 404-406: *Notably the NOAA CRW bleaching risk forecast considers DHW of 4 and 8 °C-weeks as Alert Level 1 (significant bleaching expected) and Alert Level 2 (significant bleaching and mortality expected), respectively ⁴⁸.*

L153: I understand this is what the stats say, but the green profile is behaving quite different from an ecological perspective. Supplemental figures suggest this is a profile with very few samples, which might be worth pointing out in the main text.

Response: This is very true. We have included an additional sentence to mention this point.

Lines 159-161: *Although the colony with Durusdinium spp. symbionts appeared to bleach and die faster than other colonies, with N=1 for this ITS2 profile type there was insufficient statistical power to detect whether this particular BSI trajectory differed from the rest of the population.*

References

1. Bay RA, Palumbi SR. Transcriptome predictors of coral survival and growth in a highly variable environment. *Ecology and Evolution*, 1-10 (2017).
2. Cornwell B, et al. Widespread variation in heat tolerance and symbiont load are associated with growth tradeoffs in the coral *Acropora hyacinthus* in Palau. *Elife* 10, e64790 (2021).
3. Wright RM, Mera H, Kenkel CD, Nayfa M, Bay LK, Matz MV. Positive genetic associations among fitness traits support evolvability of a reef-building coral under multiple stressors. *Global Change Biology* 25, 3294-3304 (2019).
4. Roach TN, Dilworth J, Jones AD, Quinn RA, Drury C. Metabolomic signatures of coral bleaching history. *Nature Ecology & Evolution*, 1-9 (2021).
5. Drury C, et al. Intrapopulation adaptive variance supports thermal tolerance in a reef-building coral. *Communications Biology* 5, 1-10 (2022).
6. Drury C, Dilworth J, Majerová E, Caruso C, Greer JB. Expression plasticity regulates intraspecific variation in the acclimatization potential of a reef-building coral [dataset]. Zenodo DOI:10.5281/zenodo.6877825 (2022).
7. Leggat W, Heron SF, Fordyce A, Suggett DJ, Ainsworth TD. Experiment Degree Heating Week (eDHW) as a novel metric to reconcile and validate past and future global coral bleaching studies. *Journal of Environmental Management* 301, 113919 (2022).

Reviewers' comments:

Reviewer #1 (Remarks to the Author):

The authors have addressed all of my previous comments. I believe the manuscript is now suitable for publication.

Reviewer #2 (Remarks to the Author):

Overall, the revisions have improved the manuscript. The main concern in the original MS relating to the distinction between colony growth (in terms of surface area or volume) and colony growth in terms of calcification has been resolved in the Discussion. The manuscript will be suitable for publication following revisions.

Specific comments:

Line 85: Please revise or remove. Photogrammetry is not an advancement over the buoyant weight technique as these two methods are measuring completely different things. Linear extension would perhaps be a better example, over which photogrammetry does improve upon, especially for branching corals that lack clear/visible density banding. Revise or remove.

Line 113: 5 weeks is simply not "long-term". Also on line 379. Please remove.

Line 464: This sentence doesn't make sense. Re-phrase to say, for example: "which are both able to detect changes in total CaCO₃ growth including skeletal density and secondary skeletal infilling" rather than "which are both influenced by skeletal density and secondary skeletal infilling".

Line 464: For a tank experiment, the corals have already been removed from the reef, broken into fragments, and undoubtedly moved or interfered with at various points during the experiment. This sentence about "These methods are invasive and require manipulation of the coral colony, which can potentially influence coral fitness 27" certainly holds true for in-situ field studies, but is basically irrelevant for manipulative tank experiments.

Reviewer #3 (Remarks to the Author):

The authors have done a thorough job addressing my comments and, I think, those of the other reviewers. I particularly appreciate the care they gave to expanding the methods, adding clarifying text about their experimental design and conclusions and addressing questions from R2 about the comparability of photogrammetry and calcification-based approaches. The inclusion of additional figures and analysis in the supplement has resolved my questions and concerns.

This paper will make a valuable contribution to our understanding of stress tolerance in corals.

Ford Drury

Reviewer #4 (Remarks to the Author):

The manuscript has already been reviewed once, and the authors have defended most of their changes. There are still a few issues to consider. Firstly, the Bayesian analysis undertaken by the authors is rigorous and is not the least bit controversial, especially as a simple linear model. The

technique allows an examination of the magnitude and direction of the relationships as posterior distributions, which is a highly rigorous approach. Secondly, the authors have measured linear and volumetric extension of the coral colonies, which they classify as growth. They are not necessarily the same. The first mention of the term "growth" should be qualified with a caveat, on what they actually measured, which is a linear and volumetric extension (by contrast, growth can be measured with volume displacement, or extension*density, given as calcification). Thirdly, the outliers in Figure 2B, Eggs per colony, and Total egg volume per colony may be highly influential in the analysis (Bayesian and frequentist regressions are sensitive to outliers), and removing the one outlier at the high end of the fecundity variable, in the top and bottom right-hand graphs in Figure 2B, may force the slopes in the negative direction. Please re-run the models without the outliers for colony fecundity and examine the relationships. Fourthly, while I agree that the authors show an associated tendency for individuals to tolerate heat and to "grow" (suggesting strong individuals are strong at multiple levels) I disagree with the use of the term "co-benefit" in the title and throughout the manuscript. Not showing trade-offs, or compromises, among traits is one thing, but the authors cannot assume that their alternative mechanism is co-benefit. There are positive associations between, and potentially among other, measured traits in this study, but the wording of "co-benefit" is inappropriate given the evidence at hand. Fifthly, and this is extremely minor compared with comments above, but on line 92 the authors set up a straw man. Reference 28 does not show any evidence that changes in allele frequencies are associated with a fitness cost (Ref 28 only suggests that it is possible), and on line 116, remove the word "intuitive".

Many thanks for the great ideas and insights from reviewers which we have incorporated into the presentation of our data, analysis, and discussion of the relevance of our work.

In this response to reviewers document, please note that all author responses are shown in **blue**. Quotes from the manuscript are shown in *italics*. Line numbers are shown for the simple view (tracked changes not showing). Revisions/additions compared to the last draft are shown as *underlined italics*.

Reviewers' comments:

Reviewer #1 (Remarks to the Author):

The authors have addressed all of my previous comments. I believe the manuscript is now suitable for publication.

Response: Thanks very much for your supportive comment, and for all the helpful ideas that you gave at the previous review stage.

Reviewer #2 (Remarks to the Author):

Overall, the revisions have improved the manuscript. The main concern in the original MS relating to the distinction between colony growth (in terms of surface area or volume) and colony growth in terms of calcification has been resolved in the Discussion. The manuscript will be suitable for publication following revisions.

Response: Thanks for pointing out this distinction between growth metrics. We agree that it has considerably improved the discussion.

Specific comments:

Line 85: Please revise or remove. Photogrammetry is not an advancement over the buoyant weight technique as these two methods are measuring completely different things. Linear extension would perhaps be a better example, over which photogrammetry does improve upon, especially for branching corals that lack clear/visible density banding. Revise or remove.

Response: Here we mentioned advancements in photogrammetry. We did not mean that this is an advancement over other forms of growth measurement (*e.g.*, buoyant weight or linear extension), but rather meant that as computer technology has become more advanced photogrammetry has become more accessible to people to use. To address this, we have rephrased the sentence to highlight that photogrammetry has some specific logistical advantages over other methods of determining growth since you do not even need to touch the coral (*i.e.*, non-invasive).

Lines 85-88: *Technological advances in photogrammetry now allow completely non-invasive determination of colony growth. This has some specific logistical advantages for repeated monitoring of corals in the field compared to other techniques which require removing corals from the substrate (e.g., buoyant weight) or causing potential harm (e.g., linear extension using staining)²⁷.*

Line 113: 5 weeks is simply not “long-term”. Also on line 379. Please remove.

Response: Reference 38 (McLachlan et al) is a meta-analysis of coral heat stress experiments. This study classifies short, medium, and long-term heat stress experiments, and our study falls into the category of “long-term” (> 30 days duration). Therefore, we continue with this terminology.

Line 464: This sentence doesn't make sense. Re-phrase to say, for example: "which are both able to detect changes in total CaCO₃ growth including skeletal density and secondary skeletal infilling" rather than "which are both influenced by skeletal density and secondary skeletal infilling".

Response: Done.

Line 464: For a tank experiment, the corals have already been removed from the reef, broken into fragments, and undoubtedly moved or interfered with at various points during the experiment. This sentence about "These methods are invasive and require manipulation of the coral colony, which can potentially influence coral fitness 27" certainly holds true for in-situ field studies, but is basically irrelevant for manipulative tank experiments.

Response: In this study, we removed fragments of colonies to measure certain traits (2 branches for fecundity and 6 branches for heat tolerance). However, the entire colonies remained attached to the reef where they were initially found and tagged. Therefore, our study comprises both an in-situ field study and experimental manipulations of colony fragments. We did our best to explain this in the methodology. To avoid any confusion about this we have added the word 'in situ' to the text of the methodology to clarify that the colonies themselves were not removed from the reef.

Lines 370-371: *Seventy coral colonies were tagged at 2-3 m depth and surveyed in situ repeatedly for different traits between 2017 and 2019.*

Reviewer #3 (Remarks to the Author):

The authors have done a thorough job addressing my comments and, I think, those of the other reviewers. I particularly appreciate the care they gave to expanding the methods, adding clarifying text about their experimental design and conclusions and addressing questions from R2 about the comparability of photogrammetry and calcification-based approaches. The inclusion of additional figures and analysis in the supplement has resolved my questions and concerns.

This paper will make a valuable contribution to our understanding of stress tolerance in corals.

Ford Drury

Response: Thank you for your supportive comments.

Reviewer #4 (Remarks to the Author):

The manuscript has already been reviewed once, and the authors have defended most of their changes. There are still a few issues to consider. Firstly, the Bayesian analysis undertaken by the authors is rigorous and is not the least bit controversial, especially as a simple linear model. The technique allows an examination of the magnitude and direction of the relationships as posterior distributions, which is a highly rigorous approach.

Response: Thanks for your comments on this part of our work.

Secondly, the authors have measured linear and volumetric extension of the coral colonies, which they classify as growth. They are not necessarily the same. The first mention of the term "growth" should be qualified with a caveat, on what they actually measured, which is a linear and volumetric extension (by contrast, growth can be measured with volume displacement, or extension*density, given as calcification).

Response: We agree that it would be helpful for us to qualify our definition of growth at the first appropriate instance. Therefore, we have added to the sentence in the final paragraph of the introduction the specific metrics we use in this study.

Lines 112-114: *interannual comparisons of 3D models of individual coral colonies to measure growth (change in live surface area and colony volume)*

Thirdly, the outliers in Figure 2B, Eggs per colony, and Total egg volume per colony may be highly influential in the analysis (Bayesian and frequentist regressions are sensitive to outliers), and removing the one outlier at the high end of the fecundity variable, in the top and bottom right-hand graphs in Figure 2B, may force the slopes in the negative direction. Please re-run the models without the outliers for colony fecundity and examine the relationships.

Response: Thanks for highlighting this point. We have tried rerunning the statistical test with the removal of the single most fecund colony (the same colony has the highest fecundity both in terms of eggs per colony and total egg volume, Fig 2B top and bottom panels, respectively). This did not change the trend of the regression line, with the slope of the line still not discernible from zero (i.e., a flat line or no trend, with credible intervals overlapping zero). We included this result in our manuscript and have updated the code repository to show this test.

Lines 207-208: *This trend remained the same even if the most fecund colony, a potential outlier, was removed from the analysis.*

Fourthly, while I agree that the authors show an associated tendency for individuals to tolerate heat and to “grow” (suggesting strong individuals are strong at multiple levels) I disagree with the use of the term “co-benefit” in the title and throughout the manuscript. Not showing trade-offs, or compromises, among traits is one thing, but the authors cannot assume that their alternative mechanism is co-benefit. There are positive associations between, and potentially among other, measured traits in this study, but the wording of “co-benefit” is inappropriate given the evidence at hand.

Response: Thanks for highlighting this point which had also been the viewpoint of a number of other reviewers. To address this, we have revised the title to: ‘*No apparent trade-offs associated with heat tolerance in a reef building coral*’. Furthermore, we have rephrased the term “co-benefit” throughout the manuscript using terms that highlight the positive correlations. This required changing one or two words in each case.

Lines 28-29: *Collectively, our results suggest that these corals exist on an energetic continuum where some high-performing individuals excel across multiple traits.*

Lines 62: *positive correlations across multiple traits*

Lines 188: *2.5 | Lack of trade-offs with heat tolerance and positive trait correlations*

Lines 231-232: *positive trait correlations (positive slope) or trade-offs (negative slope).*

Lines 294-295: *Further research is needed to understand if positive associations between heat tolerance and growth are also present in other coral species and over broader spatial scales.*

Lines 342-344: *Together, these results suggest that corals exist along an energetic continuum, where positive trait correlations may be derived from underlying physiological drivers like immunity⁴⁷, feeding efficiency⁵⁰, or energy storage⁵¹.*

Lines 519: *a positive slope shows positive correlations among traits.*

Fifthly, and this is extremely minor compared with comments above, but on line 92 the authors set up a straw man.

Response: We did not intend for this sentence to act as a straw man. To reduce the strength of this apparent misrepresentation of previous work, we have softened the tone of this sentence using the term “are often associated with” instead of “must be associated with”. We believe that this addresses your concern sufficiently.

Reference 28 does not show any evidence that changes in allele frequencies are associated with a fitness cost (Ref 28 only suggests that it is possible),

Response: We agree with your point here, that Ref 28 only suggest that this is the case. As such, in our text on line 92 we used this terminology (“Recent genomic evidence based on corals from contrasting thermal environments *suggests* that ...”). Therefore, we have made no further changes for this comment.

and on line 116, remove the word “intuitive”.

Response: Done